# Optimal structure learning and conditional independence testing

**Ming Gao** [1]   **Yuhao Wang** [2]   **Bryon Aragam** [1]

## Abstract

We establish a fundamental connection between optimal structure learning and optimal conditional independence testing by showing that the minimax optimal rate for structure learning problems is determined by the minimax rate for conditional independence testing in these problems. This is accomplished by establishing a general reduction between these two problems in the case of polyforests, and demonstrated by deriving optimal rates for several examples, including Bernoulli, Gaussian and nonparametric models. Furthermore, we show that the optimal algorithm in these settings is a suitable modification of the PC algorithm. This theoretical finding provides a unified framework for analyzing the statistical complexity of structure learning through the lens of minimax testing.

## 1. Introduction

Graphical models are important tools in machine learning for representing complex dependency structures that arise in a wide range of applications in causality, artificial intelligence, and statistics (Spirtes et al., 2000; Pearl, 2010; Murphy, 2012; Zhang et al., 2013). A crucial preliminary step when using graphical models is *structure learning*, which seeks to recover the underlying graph from data. The accuracy of structure learning directly impacts the performance of downstream tasks. It is well-known that structure learning is closely related to conditional independence (CI) testing, which itself is a fundamental topic that has attracted significant attention in recent years. Owing to its foundational role, CI testing has been extensively studied both methodologically and theoretically, leading to a rich body of literature on its statistical properties (Zhang et al., 2011; Shah & Peters, 2020; Canonne et al., 2018; Neykov et al., 2021). In particular, structure learning methods often rely

[1]Booth School of Business, University of Chicago [2]Amazon. Correspondence to: Ming Gao <minggao@chicagobooth.edu>.

*Proceedings of the 43rd International Conference on Machine Learning*, Seoul, South Korea. PMLR 306, 2026. Copyright 2026 by the author(s).

on CI testing as a subroutine for inferring graphical dependencies (Spirtes & Glymour, 1991; Maathuis et al., 2018).

The relationship between CI testing and structure learning has long been understood from the algorithmic perspective, which takes CI tests as black-box oracles and often assumes perfect CI information to guarantee graph recovery. However, the relationship between their finite-sample complexity and statistical hardness has yet to be formalized. Specifically, given that graphical models are intrinsically defined by the Markov property, a deeper understanding of how the statistical difficulty of CI testing governs the learning of graphical structures is both natural and fundamental.

In undirected graphical models (Markov random fields), the information-theoretic limits of structure learning—specifically, the minimum number of samples required for reliable graph recovery—have been widely investigated (Wang et al., 2010; Drton & Maathuis, 2017; Misra et al., 2020). For directed acyclic graphs (DAGs), however, much of the optimality literature has focused on the Gaussian setting, where the analysis exploits strong distributional properties of the Gaussian. As a result, existing optimality results do not readily extend to discrete or nonparametric settings. By contrast, CI testing in its own right has been studied more broadly, with well-established minimax results across various distributional settings (Canonne et al., 2018; Neykov et al., 2021). Moreover, existing optimality results do not establish a general (information-theoretic) reduction between structure learning and CI testing, which would be useful both in its own right, as well as for future investigations into the hardness of structure learning broadly. These gaps suggest an opportunity to leverage statistical results in CI testing to better understand the sample complexity of structure learning for DAGs in general.

In this paper, we establish a general reduction between the minimax optimality of structure learning in DAG models and the minimax optimality of CI testing. While there is no such reduction even for Gaussian models, existing optimality results (Wang et al., 2024; Daskalakis et al., 2025) implicitly exploit this connection. In addition to establishing this reduction in a general setting that allows us to analyze non-Gaussian models, we also show how existing Gaussian results can be deduced as a special case. We focus on poly-forests, a tractable yet rich subclass of DAGs that

captures structured dependencies (Chow & Liu, 1968; Dasgupta, 1999), serving as a principled starting point toward fully general DAGs. Although optimality for learning general DAGs has been studied (Gao et al., 2022), specific distributional assumptions are needed in the analysis (e.g. linear SEM with equal error variances). We develop results under generic conditions that apply to both parametric and nonparametric distributions. Our work provides a framework connecting the statistical complexity of CI testing with that of structure learning, offering new insights and understanding of sample-efficient DAG recovery in diverse settings.

## 1.1. Contributions

Our contributions in this work are threefold:

1. We establish a connection (Theorem 3.1) between the statistical complexity of conditional independence testing and structure learning problems. We show that the minimax optimal sample complexity of learning any poly-forest is

$$n \asymp \frac{\log d}{c^\alpha}, \qquad (1)$$

when the minimax optimal testing radius of the corresponding CI testing problem is $n^{-1/\alpha}$ for some $\alpha > 0$ depending on the modeling setup, and $d$ is the number of nodes, $c$ is a parameter that captures the signal strength (cf. Section 2).

2. We apply this general result to derive the minimax optimal sample complexity for several practical examples, including Bernoulli, Gaussian, and nonparametric continuous distributions, and discuss their differences. We also show that the optimal sample complexity is achieved by an efficient algorithm based on the classical PC algorithm.

3. We conduct experiments to verify our theoretical findings for structure learning using Bernoulli, Gaussian, and nonparametric continuous distributions. These empirical results confirm that a powerful CI test integrated into the PC-type algorithm leads to consistent and accurate structure recovery.

While it may not come as a surprise that there is *some* connection between CI testing structure learning, the profound simplicity of (1) merits some pause. The reduction depends only on the dimension $d$, the signal $c$, and the minimax exponent $\alpha$ of CI testing. Moreover, this dependence is both simple and mild (i.e. only logarithmic in $d$), and applies to any model for which the minimax CI testing radius can be written down.

## 1.2. Related work

**Conditional independence testing**    CI testing forms a crucial building block in machine learning and reasoning tasks. Given a triplet of random variables $(X, Y, Z)$, the problem asks whether $X$ is conditionally independent of $Y$ given $Z$. Numerous methods (e.g. Dawid, 1979; Fukumizu et al., 2007; Ramsey, 2014; Li & Fan, 2020) have been developed to address this fundamental problem. One prominent class of methods relies on measuring the distance between the conditional distribution $P(X, Y \mid Z)$ and the product of the marginals $P(X \mid Z)P(Y \mid Z)$. For Gaussian variables, the problem reduces to testing whether the partial correlation is zero (Fisher, 1915). For discrete variables, hypothesis tests based on chi-squared statistics are widely employed (Ireland & Kullback, 1968; Darroch et al., 1980). Recent advancements have also explored kernel-based methods (Gretton et al., 2007; Zhang et al., 2011) and information-theoretic measures such as conditional mutual information (Runge, 2018; Berrett & Samworth, 2019) that can capture complex nonlinear dependencies in a nonparametric fashion.

From the theoretical standpoint, CI testing has been investigated from the perspective of minimax optimality. For discrete distributions, the problem is well-understood and the minimax optimal rates have been established (Chan et al., 2014; Diakonikolas & Kane, 2016; Canonne et al., 2018). In the nonparametric setting, Neykov et al. (2021) studied the fundamental limits of CI testing, deriving minimax optimal rates under smoothness assumptions. In addition, practical CI tests have been proposed (Kim et al., 2022; 2024). In particular, Jamshidi et al. (2024) devised a Von Mises estimator for mutual information as a valid CI test under smoothness conditions. These theoretical results provide valuable insights into the inherent difficulty of CI testing under different distributional setups.

**Graphical model structure learning**    For undirected graphical models, the structure learning problem reduces to support recovery of the precision matrix (Meinshausen & Bühlmann, 2006; Friedman et al., 2008; Cai et al., 2011; Liu et al., 2009). For DAG learning, approaches can be broadly categorized into three main paradigms: score-based methods (e.g. greedy equivalence search) (Chickering, 2002; Nandy et al., 2018), constraint-based methods (e.g. PC algorithm) (Spirtes & Glymour, 1991; Friedman et al., 2013), and methods that leverage specific distributional assumptions (Shimizu et al., 2006; Hoyer et al., 2008; Peters et al., 2011; 2014; Peters & Bühlmann, 2014). Especially, constraint-based methods rely on valid CI tests and the faithfulness assumption to operate (Kalisch & Bühlman, 2007; Marx et al., 2021), thus development of reliable and efficient CI tests directly impacts the performance of these structure learning methods. For tree-structured models, the Chow-Liu algorithm (Chow & Liu, 1968; Chow & Wagner, 1973)

provides an efficient method to find the optimal undirected tree structure by constructing a maximum weight spanning tree based on pairwise mutual information. Learning poly-trees/forests is generally more complex than learning undirected trees models, while remaining tractable compared to learning general DAGs (Rebane, 1987; Srebro, 2003; Tan et al., 2010; 2011). This problem has seen developments recently (Gao & Aragam, 2021; Azadkia et al., 2021; Jakobsen et al., 2022). Furthermore, learning poly-trees/forests is relevant in various applications, including reasoning and density estimation (Kim & Pearl, 1983; Liu et al., 2011).

**Sample complexity of structure learning** For undirected graphical models, optimal sample complexities have been derived for various classes of undirected graphs, such as sparse graphs with bounded degree (Wang et al., 2010; Santhanam & Wainwright, 2012; Bresler, 2015; Vuffray et al., 2016; Abbe, 2018; Misra et al., 2020). In contrast, determining the sample complexity for DAG learning is considerably more challenging due to the inherent asymmetry and the larger space of possible graph structures. Ghoshal & Honorio (2017) provided lower bounds for structure learning of general DAG models. Gao et al. (2022) established matching upper and lower bounds on the sample complexity as $\Theta[q \log(d/q)]$ for learning Gaussian DAGs under the restrictive equal variance assumption (Peters & Bühlmann, 2014; Loh & Bühlmann, 2014; Chen et al., 2019), where $q$ is the degree of the DAG, and this optimality result has been further extended to linear dynamic models (Veedu et al., 2024). Wang et al. (2024) concluded the optimal sample complexity as $\Theta(\log d/c^2)$ for learning Gaussian poly-trees, where $c$ is the faithfulness parameter, and provided an efficient algorithm based on the classic PC algorithm (Spirtes et al., 2000). For nonparametric models, Jamshidi et al. (2024) applied the devised Von Mises CI test to the PC algorithm and obtain a dependence of $\mathcal{O}[(\frac{\Delta}{I_{\min}} \log d)^2]$, where $\Delta$ is the graph degree and $I_{\min}$ is a lower bound on the conditional mutual information, assuming the density function is sufficiently smooth and lower bounded (from zero). In contrast, our work focuses on establishing a general connection between minimax CI testing and structure learning, including method-agnostic minimax lower bounds, that apply broadly to general distributions (especially the nonparametric ones) with minimal assumptions.

## 2. Preliminaries

Given a directed acyclic graph (DAG) $G = (V, E)$, $\mathrm{pa}(k) = \{j : (j, k) \in E\}$ denotes the parents of node $k \in V$. The skeleton of $G$, $\mathrm{sk}(G)$, is the undirected graph formed by removing directions of all the edges in $G$. A triplet $(j, \ell, k)$ is called unshielded if both $j, k$ are adjacent to $\ell$ but not adjacent to each other, graphically $j - \ell - k$; and is called a $v$-structure if additionally $j, k$ are parents of $\ell$, i.e. $j \rightarrow \ell \leftarrow k$.

A path is a sequence of distinct nodes $(h_1, \ldots, h_\ell)$ such that $(h_j, h_{j+1})$ is in $\mathrm{sk}(G)$. A forest is an undirected graph where any two nodes are connected by at most one path. A poly-forest is a DAG whose skeleton is a forest. Denote the set of all poly-forests over $d$ nodes as $\mathcal{T} = \mathcal{T}_d$.

A distribution $p$ satisfies the Markov property with respect to a DAG $G$ with $d$ nodes if the following factorization holds

$$p(X) = p(X_1, \ldots, X_d) = \prod_{k=1}^{d} p(X_k \mid \mathrm{pa}(k))$$

We consider the problem of *structure learning*, with a focus on poly-forests, where we assume the existence of a poly-forest $G \in \mathcal{T}$ such that $p$ is Markov to $G$, and we aim to recover $G$ given $n$ i.i.d. samples from $p$. In general, it is well-known that the DAG is not identifiable from observational data alone. Assuming faithfulness, $G$ is identified up to its Markov equivalence class (MEC), which is the set of DAGs that encode the same set of conditional independencies as $G$ and is represented by completed partially directed acyclic graph (CPDAG), denoted by $\overline{G}$. We refer the readers to Koller & Friedman (2009) for more preliminaries on graphical models.

### 2.1. Measuring dependence in general models

Since we assume the underlying DAG belongs to the class of poly-forests, the usual faithfulness assumption can be relaxed to tree-faithfulness (Wang et al., 2024) for successful recovery. To derive uniform, finite-sample bounds, we also need conditions on the minimum signal strength, as is standard in model selection and testing literature. We first define a general notion of dependence measure that will be used to quantify the signal strength.

**Definition 2.1** (Dependence measure). Let $(X, Y, Z) \sim p$ be a triplet of random variables subject to some distribution $p$. A *(conditional) dependence measure* $m(X; Y \mid Z)$ is a functional of $p$ such that (1) $m(X; Y \mid Z) \geq 0$; and (2) $m(X; Y \mid Z) = 0$ if and only if $X \perp\!\!\!\perp Y \mid Z$ under $p$.

If $Z = \emptyset$, we simplify the notation by writing the *marginal dependence* as $m(X; Y) = m(X; Y \mid \emptyset)$. This general measure of dependence is used to simplify our main theorem statement; in our examples (Sections 4-5), we specify the dependence measure as the usual correlation coefficient for Gaussian distributions, total variation with the product of marginals for Bernoulli distributions, and total variation distance to the nearest independent instance for nonparametric continuous distributions (Canonne et al., 2018; Neykov et al., 2021; Wang et al., 2024). In more general settings (e.g. user-defined models), the dependence measure can be chosen based on modeling preference, as long as there exist consistent or efficient CI tests achieving provable error guarantees, it can be embedded into our framework. Us-

ing this dependence measure, we can now define strong tree-faithfulness in a generic form.

**Definition 2.2** ($c$-strong tree-faithfulness). A distribution $p$ is $c$-strong tree-faithful to a poly-forest $G$ with respect to the dependence measure $m$ if

1. For any two nodes connected $j - k$, we have $m(X_k; X_j \mid X_\ell) \geq c$ for $\ell \in V \cup \{\emptyset\} \setminus \{k, j\}$;

2. For any $v$-structure $k \rightarrow \ell \leftarrow j$, we have $m(X_k; X_j \mid X_\ell) \geq c$.

Tree-faithfulness is a relaxed version of the general faithfulness assumption, which requires the CI relationships in data distribution to reflect the existence of edges in graph (Koller & Friedman, 2009; Wang et al., 2024).

### 2.2. CI testing and structure learning

Since we aim to establish a statistical connection between CI testing and structure learning, we define each problem and the associated statistical quantities of interest as follows. In particular, given the close relationship between these two problems and the fact that structure learning often relies on CI testing as a subroutine, we first introduce the CI testing problem before defining the poly-forest learning problem. To ground the abstract formulation of CI testing, we begin with a concrete example under the nonparametric setting to highlight the key concepts and notations. See Section 5 and Appendix G for more details of this example.

**Example 1** (Nonparametric models). *Let $\mathcal{P}$ be all continuous distributions supported on $[0, 1]^3$ that admit Lipschitz continuous densities $p$. For each distribution $p \in \mathcal{P}$ over a triplet of variables $(X, Y, Z)$, we measure the dependence between $X$ and $Y$ given $Z$ by the total variation distance $\inf_{q \in \mathcal{Q}} \|p - q\|_1$, where $\mathcal{Q}$ is the set of all conditionally independent distributions in $\mathcal{P}$. This serves as a valid dependence measure $m$ for nonparametric distributions. The nonparametric CI testing problem asks for a test to distinguish the following two hypotheses:*

$$\mathcal{H}_0: \quad p(X, Y, Z) \in \mathcal{P} \quad s.t. \quad X \perp\!\!\!\perp Y \mid Z$$
$$\mathcal{H}_1: \quad p(X, Y, Z) \in \mathcal{P} \quad s.t. \quad \inf_{q \in \mathcal{Q}} \|p - q\|_1 \geq r,$$

*for some $r > 0$. A sufficiently large $r$ is necessary for consistent testing (Shah & Peters, 2020), and is also used to study the statistical hardness in terms of the minimax testing radius (defined in the sequel).*

Building on the intuition from Example 1, we formally introduce the CI testing problem in a generic form, which can be instantiated under various distributional settings. Based on the elements of CI testing, we will then proceed to define the poly-forest learning problem such that the relationship

between the two problems is explicit and clear. At its core, CI testing is a fundamental statistical problem concerned with distinguishing distributions over triplet of variables $(X, Y, Z)$.

**Definition 2.3** (Conditional independence testing). A conditional independence testing problem $\mathcal{C}(\mathcal{P}, m, n)$ is defined by a class of distributions $\mathcal{P}$, dependence measure $m$ and sample size $n$, and aims to distinguish two hypotheses of distributions:

$$\mathcal{H}_0: \quad p(X, Y, Z) \quad \text{s.t. } m(X; Y \mid Z) = 0 \Leftrightarrow X \perp\!\!\!\perp Y \mid Z$$
$$\mathcal{H}_1: \quad p(X, Y, Z) \quad \text{s.t. } m(X; Y \mid Z) \geq r$$

where $p \in \mathcal{P}$, and $r$ is the signal strength to distinguish the two hypotheses. A conditional independence test $\psi$ is a function that takes $n$ i.i.d. samples from $p$ and outputs a binary decision, i.e. $\psi : \{(X^{(i)}, Y^{(i)}, Z^{(i)})\}_{i=1}^n \mapsto \{0, 1\}$ where $\psi = 0$ indicates selecting the null $\mathcal{H}_0$. Fixing the sample size $n$, the minimax optimal testing radius of $\mathcal{C}(\mathcal{P}, m, n)$ is the infimum of $r = r_n > 0$ in terms of $n$ (up to constants) such that there exists a test whose Type-I and Type-II errors are controlled:

$$\inf_\psi \left\{ \sup_{p \in \mathcal{H}_0} \mathbb{E}_p[\psi] + \sup_{p \in \mathcal{H}_1} \mathbb{E}_p[1 - \psi] \right\} \leq \frac{1}{10}.$$

For a certain CI testing problem, one needs to specify the model class $\mathcal{P}$, and the dependence measure $m$. Consequently, the hypothesis classes $\mathcal{H}_0, \mathcal{H}_1$ are determined. In Example 1, $\mathcal{P}$ includes all the Lipschitz densities over $[0, 1]^3$ and $m$ is given by the total variational distance. One goal of studying $\mathcal{C}(\mathcal{P}, m, n)$ is to derive the minimax testing radius $r_n$ such that the conditionally dependent and independent instances can be distinguished accurately. We proceed to introduce the poly-forest learning problem using this set-up.

**Definition 2.4** (Poly-forest learning problem). A poly-forest learning problem $\mathcal{F}(\mathcal{P}, m, c)$ is defined by the following family of distributions (i.e. graphical model):

$$\mathcal{F} = \Big\{ (p, G) \Big| \ p \text{ is Markov and } c\text{-strong tree-faithful}$$
$$\text{to } G \in \mathcal{T} \text{ with respect to } m,$$
$$\text{and } \forall j, k, \ell \in [d], p_{X_j, X_k, X_\ell} \in \mathcal{P} \Big\}.$$

The goal is to learn the Markov equivalence class of $G$ given i.i.d. samples from $p$. The optimal sample complexity of $\mathcal{F}(\mathcal{P}, m, c)$ refers to the smallest integer $n$ as sample size in terms of the number of nodes $d$ and the strong tree-faithfulness parameter $c$ (up to constants) such that the graph structure can be confidently learned by some estimator $\widehat{G}$:

$$n(\mathcal{F}) = \inf \Big\{ n \ \Big| \ \exists \widehat{G} \text{ s.t. } \sup_{(p, G) \in \mathcal{F}} \mathbb{P}(\widehat{G} \neq \overline{G}) \leq \frac{1}{10} \Big\}.$$

The poly-forest model is defined by requiring the marginal distributions of all node triplets to belong to $\mathcal{P}$, thereby sharing the distributional properties considered in the CI testing problem $\mathcal{C}(\mathcal{P}, m, n)$, e.g. Gaussianity or smoothness. Combined with Markov property and $c$-strong tree-faithfulness (with respect to $m$), this essentially implies that every node triplet comes from either $\mathcal{H}_0$ or $\mathcal{H}_1$, with the signal strength $r$ in the testing problem replaced by the faithfulness parameter $c$ (and the two notations would be used interchangeably whenever the context is clear). We focus on the optimal sample complexity of recovering the CPDAG for the poly-forest in $\mathcal{F}(\mathcal{P}, m, c)$. The number $1/10$ in both Definitions 2.3 and 2.4 can be replaced with any fixed small constant independent with other problem parameters, e.g. 0.05, as its dependence is not of primary interest.

The estimator that achieves the optimal sample complexity for $\mathcal{F}(\mathcal{P}, m, c)$ turns out to be the well-known PC algorithm (Spirtes & Glymour, 1991), and establishing that this is optimal in general is interesting in and of its own right. More precisely, we adapt the PC algorithm to poly-forests, as done in Wang et al. (2024) for Gaussian distributions. The resulting algorithm is called `PC-tree`, and determines the edges between any two nodes by testing their (conditional) independence given one other node. The algorithm takes a valid CI test as input, is efficient and runs in polynomial time in number of nodes $d$, and is readily extended for poly-forests for general distributions. We detail the specifics of the `PC-tree` algorithm in Appendix A.

## 3. Equivalence between CI testing and structure learning

We present our main result below, establishing a fundamental connection between CI testing $\mathcal{C}(\mathcal{P}, m, n)$ and structure learning $\mathcal{F}(\mathcal{P}, m, c)$.

**Theorem 3.1.** *Given a conditional independence testing problem $\mathcal{C}(\mathcal{P}, m, n)$ with an optimal test $\psi$ achieving the minimax testing radius $r_n \asymp n^{-1/\alpha}$, if there exist hard instances $p_0 \in \mathcal{H}_0$ and $p_1 \in \mathcal{H}_1$ that are Markov and $c$-strong tree-faithful, then the optimal sample complexity of learning $\mathcal{F}(\mathcal{P}, m, c)$ is*

$$n \asymp \frac{\log d}{c^\alpha}, \tag{2}$$

*which is achieved by `PC-tree` with $\psi$.*

Theorem 3.1 establishes a fundamental statistical reduction between the two problems: the optimality of a CI testing problem implies the optimality of corresponding structure learning for poly-forests. In particular, the inherent difficulty in CI testing directly translates to the difficulty in poly-forest learning, and once the minimax solution (an optimal CI test $\psi$) is obtained, it can be directly plug-in for learning the poly-forest.

The condition in Theorem 3.1 looks for two hard instances $p_0 \in \mathcal{H}_0$ and $p_1 \in \mathcal{H}_1$ that are difficult to distinguish in the sense of small KL-divergence, see (C2) for the detailed requirement. We stress that this condition is imposed merely on triplet of variables (cf. Definition 2.3), and is a standard step to establish lower bounds when studying the minimax optimality for CI testing. Additional requirements on the graphical properties, i.e. Markov and strong tree-faithfulness, of these instances are introduced to ensure their validity in the context of poly-forest learning. As will be shown in Section 5, this technical requirement sometimes necessitates minor modifications on existing constructions in the literature of CI testing.

In the optimal sample complexity of structure learning (2), the effect of dimensionality comes in as a factor of $\log d$, and the exponent $\alpha$ on dependence of signal draws connection between CI testing and structure learning problems, as it directly translates from the optimal testing radius to the optimal sample complexity. For parametric distributions, we typically have $\alpha = 2$, corresponding to the parametric rate. While for nonparametric distributions, it is often the case that $\alpha > 2$ and depends on the smoothness conditions. We provide examples on both cases in Sections 4-5.

To clarify our contribution relative to the existing literature, we compare with the optimality result in Wang et al. (2024), which crucially relies on the specific properties of the Gaussian distribution and therefore does not extend to discrete or nonparametric models. In contrast, our black-box reduction between testing and structure learning is model-agnostic and applicable to general, including fully nonparametric, settings where partial correlation is no longer useful and linearity fails. It allows us to obtain minimax-optimal sample complexity guarantees without relying on specific parametric properties of the Gaussian distribution. Moreover, such reduction result highlights the fundamental connection with CI testing in the view of sample efficiency.

To apply Theorem 3.1 on concrete problems to obtain optimality, one needs to specify the distributional assumptions in the CI testing problem, design an optimal CI test, and find proper instances from the (in)dependent hypotheses that are close enough in KL-divergence. In the remainder of this paper, we show how to apply Theorem 3.1 to Bernoulli, Gaussian, and nonparametric continuous distributions. In practice, given a powerful test $\psi$ tailored to the models satisfying certain distributional assumptions, one can directly integrate it into `PC-tree` algorithm for efficient poly-forest learning. We demonstrate this across the three distributional settings in the experiments in Section 6.

A full detailed statement of Theorem 3.1 with explicit constants, technical conditions, additional discussions, and proof is provided in Theorem C.1 in Appendix C. We end this section with a proof sketch: For the upper bound, we

first construct an amplified version of the test $\psi$ via the median trick (Motwani, 1995), which translates the constant Type I and II error guarantees into an exponentially decaying error probability (cf. Proposition C.2). We then apply this amplified test as a subroutine to PC-tree. The desired upper bound follows by adapting the analysis in the proof of Theorem 4.3 in Wang et al. (2024), noting that the error probability of each CI test is sufficiently controlled to ensure uniform consistency over all edge decisions. For the lower bound, we apply Tsybakov's method (Corollary H.4). We construct an ensemble of $\asymp d$ candidate distributions, each corresponding to a distinct poly-forest, by embedding the hard instances $p_0$ and $p_1$ into disjoint three-node subgraphs and stacking them across the graph. The lower bound follows by verifying that the pairwise KL divergences between these distributions are uniformly bounded.

## 4. Applications to Bernoulli and Gaussian distributions

In this and the following sections, we illustrate by examples the broad applicability of the main optimality result (Theorem 3.1) through a series of representative distributions. For each example, we specify the associated model class, provide a valid CI test, and give hard instances that satisfy the condition of Theorem 3.1. For the sake of space, we state and discuss the key conclusions in the main paper and defer full details of the model class descriptions to Appendix B.

### 4.1. Bernoulli distribution

We start with Bernoulli distribution. As mentioned, it suffices to define the CI testing problem to set the stage. We consider all multivariate Bernoulli distributions with dimension being three, i.e. $\mathbb{P}(X = x, Y = y, Z = z) = p(x, y, z), \forall (x, y, z) \in \{0, 1\}^3$ and $\sum_{x,y,z} p(x, y, z) = 1$. Formal details are deferred to Appendix B.1. We measure dependence using

$$m^{Ber}(X, Y \mid Z)$$
$$= \sum_z p(z) \sum_{x,y} \left| p(x, y \mid z) - p(x \mid z) p(y \mid z) \right|$$

which is the total variation distance with the product of its marginals and is a valid dependence measure. These elements lead to CI testing problem $\mathcal{C}^{Ber}$ and the poly-forest learning problem $\mathcal{F}^{Ber}$, where $Ber$ denotes "Bernoulli". $\mathcal{F}^{Ber}$ effectively contains all multivariate Bernoulli distributions with dimension $d$ and Markov and $c$-strong tree-faithful to some poly-forest $G \in \mathcal{T}$:

$$(X_1, X_2 \ldots, X_d) \sim \mathbb{P}(X_1 = x_1, X_2 = x_2, \ldots, X_d = x_d),$$
$$\forall (x_1, x_2 \ldots, x_d) \in \{0, 1\}^d.$$

Now we proceed to construct a test based on thresholding

the estimation of the dependence measure by the sample counterpart, inspired by the classical $\chi^2$ independence test:

$$\widehat{m}^{Ber}(X, Y \mid Z) \tag{3}$$
$$= \sum_z \widehat{p}(z) \sum_{x,y} \left| \widehat{p}(x, y \mid z) - \widehat{p}(x \mid z) \widehat{p}(y \mid z) \right|$$
$$\psi^{Ber} = \mathbb{1}\{\widehat{m}^{Ber}(X, Y \mid Z) \geq c/2\} \tag{4}$$

where $\widehat{p}(x) = \sum_i \mathbb{1}\{X_i = x\}/n$ for $x \in \{0, 1\}$ and other estimates are analogously defined.

For the hard instance constructions, we consider $p_0^{Ber}(X, Y, Z)$ under which $X, Y, Z$ are independent $\text{Bern}(\frac{1}{2})$ random variables, and the alternative distribution $p_1^{Ber}(X, Y, Z)$ is given as follows:

$$Z \sim \text{Bern}\left(\frac{1}{2}\right), \quad X \mid Z \sim \begin{cases} \text{Bern}(\frac{1}{2} + c) & Z = 1 \\ \text{Bern}(\frac{1}{2} - c) & Z = 0 \end{cases},$$
$$Y \mid X \sim \begin{cases} \text{Bern}(\frac{1}{2} + c) & X = 1 \\ \text{Bern}(\frac{1}{2} - c) & X = 0 \end{cases}.$$

It is easy to see that $p_0^{Ber}$ is constructed to be Markov and tree-faithful to an empty graph, while $p_1^{Ber}$ is constructed to follow a three-node chain graph $Z \to X \to Y$, both of which are poly-forests.

Then the following theorem checks the validity of the test and construction, and concludes the optimality of learning Bernoulli poly-forest via Theorem 3.1. See proof in Appendix D.

**Theorem 4.1.** $\psi^{Ber}$ *is optimal for* $\mathcal{C}^{Ber}$ *with optimal testing radius being* $r_n \asymp n^{-1/2}$, *and the optimal sample complexity of learning* $\mathcal{F}^{Ber}$ *is*

$$n \asymp \frac{\log d}{c^2},$$

*which is achieved by* PC-tree *with* $\psi^{Ber}$.

### 4.2. Gaussian distribution

We next consider Gaussian distribution as another parametric example to illustrate the main result. Since this application will recover the established optimality in Wang et al. (2024), we only collect the key elements and defer most details to Appendix B.2. We consider all multivariate Gaussian distributions with dimension being three and measure dependence using the partial correlation coefficient (cf. (6)). The Gaussian CI testing problem $\mathcal{C}^{Gauss}$ and the Gaussian poly-forest learning problem $\mathcal{F}^{Gauss}$ are defined, where $Gauss$ represents "Gaussian".

We employ a valid CI test $\psi^{Gauss}$ similar to $\psi^{Ber}$ by thresholding the sample partial correlation (cf. (7)). For the hard instances, let $p_0^{Gauss}$ be $\mathcal{N}(\mathbf{0}_3, I_3)$ and $p_1^{Gauss}$ be generated

as

$$Z \sim \mathcal{N}(0,1), \quad X = \beta Z + \mathcal{N}(0, 1 - \beta^2),$$
$$Y = \beta X + \mathcal{N}(0, 1 - \beta^2),$$

where $\beta = 2c$. Likewise in the Bernoulli case, $p_0^{Gauss}$ is constructed to be Markov and tree-faithful to an empty graph, and $p_1^{Gauss}$ is constructed for the chain graph $Z \rightarrow X \rightarrow Y$. It remains to check the validity of $\psi^{Gauss}$, $p_0^{Gauss}$ and $p_1^{Gauss}$. Applying Theorem 3.1, we have the optimality of learning Gaussian poly-forest (proof is in Appendix E):

**Theorem 4.2.** $\psi^{Gauss}$ *is optimal for* $\mathcal{C}^{Gauss}$ *with optimal testing radius being* $r_n \asymp n^{-1/2}$, *and the optimal sample complexity of learning* $\mathcal{F}^{Gauss}$ *is* $n \asymp \log d/c^2$, *which is achieved by* PC-tree *with* $\psi^{Gauss}$.

### 4.3. Extension to optimal poly-tree learning

Poly-tree is a subset of poly-forest whose skeleton has any two nodes be connected by *exactly* one path. The optimality results for both Bernoulli and Gaussian distributions can be extended to poly-tree learning, whose problem definition simply replaces the poly-forest in Definition 2.4 by poly-tree. Since poly-tree is a sub-class of poly-forest, the upper bound result of PC-tree applies. Then it suffices to derive a lower bound to match the $\log d/c^2$ dependence, which requires the construction to be Markov and faithful to not just poly-forest, but further poly-tree.

The optimality of poly-tree learning in Gaussian setting is investigated in Wang et al. (2024), where the optimal sample complexity is shown to be $n \asymp \log d/c^2$ with the same notion of $c$-strong tree-faithfulness assumed (Definition 2.2). Here we show the same extension holds for Bernoulli poly-tree due to the parametric nature. The main difficulty lies in the lower bound construction, for which we cannot directly adopt the construction in Wang et al. (2024) due to the discrepancy between continuous and discrete distributions. While analogously, we consider all possible directed Markov chains in form of $X_{\pi_1} \rightarrow X_{\pi_2} \rightarrow \ldots X_{\pi_d}$ for some permutation $\pi$, which is a subset of poly-trees, and specify the construction as follows: For each Markov chain $T$, consider $P_T$ given by $X_{\pi_1} \sim \text{Bern}(1/2)$ and for $k = 2, 3, \ldots, d$,

$$X_{\tau_k} \mid X_{\pi_{k-1}} \sim \begin{cases} \text{Bern}(\frac{1}{2} + c) & X_{\pi_{k-1}} = 1 \\ \text{Bern}(\frac{1}{2} - c) & X_{\pi_{k-1}} = 0 \end{cases}. \quad (5)$$

The idea is to add small perturbation according to the amount of faithfulness parameter such that the KL divergence between distributions induced by different Markov chains can be bounded. The validity of this construction is due to Lemma F.1 and F.2 proved in Appendix F. We conclude the optimality below:

**Theorem 4.3.** *The optimal sample complexity of Bernoulli or Gaussian poly-tree learning is* $n \asymp \log d/c^2$, *which is achieved by* PC-tree *with* $\psi^B$ *or* $\psi^G$.

## 5. Application to nonparametric models

For both Bernoulli and Gaussian distributions, the optimal sample complexity for learning poly-forest is $n \asymp \log d/c^2$, i.e. $\alpha = 2$ in Theorem 3.1. This result primarily arises from the parametric nature of these distributions. To explore the implications beyond the parametric setting, we now shift our focus to a nonparametric continuous distribution. This allows us to examine how the nonparametric nature influences the value of $\alpha$, leading to a distinct theoretical outcome.

We adopt the framework in Neykov et al. (2021) and provide the essential elements in Appendix B.3, see details therein. As alluded to in Example 1, we consider all continuous distributions over $[0,1]^3$, which admit continuous densities $p(X, Y, Z)$. We measure the dependence using

$$m^{NP}(X, Y \mid Z) = \inf_{q \in \mathcal{Q}} \|p - q\|_1,$$

where $\mathcal{Q}$ is the class of continuous distributions $q(X, Y, Z)$ over $[0,1]^3$ such that $X \perp\!\!\!\perp Y \mid Z$, and $\|p - q\|_1 = \int |p(x, y, z) - q(x, y, z)| dx dy dz$. This measures the distance to the closest (conditionally) independent distributions. In addition, smoothness conditions are imposed on the densities (cf. Definitions B.1-B.2), characterized by a smoothness parameter $s$ and used to contrast with the parametric cases. Together, these defines the CI testing class $\mathcal{C}^{NP}$ and the poly-forest learning problem $\mathcal{F}^{NP}$, where $NP$ denotes "nonparametric".

Now we proceed to verify the applicability of Theorem 3.1 for this nonparametric setting. We use the minimax optimal CI test introduced in Section 5.3 of Neykov et al. (2021), denoted as $\psi^{NP}$, which is based on classic U-statistics to measure the (conditional) dependence between the $X$ and $Y$. $\psi^{NP}$ is a valid choice and satisfies the type I and II error controls with $\alpha = \frac{5s+2}{2s}$ (see Theorem 5.6 therein).

For the hard instances, we design the constructions, denoted as $p_0^{NP}$ and $p_1^{NP}$, based on a modification of the ones used for proving lower bounds for nonparametric CI testing in Neykov et al. (2021). Although the original constructions of $p_0$ and $p_1$ are close in KL divergence, the issue lies in the unfaithfulness of the distribution, thus they cannot be directly applied for poly-forest learning setting. Specifically, the original construction of the alternative $p_1$ only characterizes the conditional dependence but accidentally leads to marginal independence between variables, which cannot be faithful to any poly-forest (over triplet of nodes).

We modify the construction by adding back the marginal dependence with extra complexity in the analysis. Specifically,

let $p_0^{NP}$ be independent uniform distributions $Unif^3[0,1]$, which is Markov to the empty graph. For $p_1^{NP}$, we consider a $V$-structure $X \to Z \leftarrow Y$, which is a three-node polyforest. Under this graph, a faithful distribution is supposed to have $X \perp\!\!\!\perp Y$ while $X \not\perp\!\!\!\perp Y \mid Z$ and $X \not\perp\!\!\!\perp Z, Y \not\perp\!\!\!\perp Z$. Thus $p_1^{NP}$ is specified as follows: $X, Y \sim Unif[0,1]$ and $p_{Z \mid X, Y}$ is given by a perturbation to the uniform depending on $X$ and $Y$:

$$p(Z \mid X, Y) = 1 + \gamma_\Delta(X, Y)\eta_\nu(Z),$$

for some functions $\gamma_\Delta$ and $\eta_\nu$ governing the perturbation, which are specified in the proof. This construction is in the same form as the one in Neykov et al. (2021). However, the functions are constructed such that the $X$ (or $Y$) and $Z$ are marginally dependent, thus faithfulness is satisfied. See more details about these constructions in the proof (Appendix G). Applying Theorem 3.1, we have the optimality of learning nonparametric continuous poly-forest.

**Theorem 5.1.** $\psi^{NP}$ *is optimal for* $\mathcal{C}^{NP}$ *with optimal testing radius being* $r_n \asymp n^{-2s/(5s+2)}$, *and the optimal sample complexity of learning* $\mathcal{F}^{NP}$ *is*

$$n \asymp \frac{\log d}{c^{\frac{5s+2}{2s}}},$$

*which is achieved by* `PC-tree` *with* $\psi^{NP}$.

Theorem 5.1 highlights the larger sample complexity of nonparametric setting compared to the parametric one for poly-forest learning. The difference lies in the dependence on the strong tree-faithfulness parameter $c$, where the resulting exponent $\alpha = \frac{5s+2}{2s} > 2$ comes directly from the intrinsic hardness of nonparametric CI testing. Moreover, the modification for $p_0^{NP}$ and $p_1^{NP}$ illustrates the additional complexity in the analysis to ensure faithfulness.

## 6. Experiments

We conducted experiments to validate our theoretical findings on CI testing and structure learning with `PC-tree` algorithm (detailed in Algorithm 1). Specifically, we conducted a series of simulation studies for structure learning in Bernoulli, Gaussian, and nonparametric continuous distributions, corresponding to the examples provided in Sections 4-5. We simulate poly-forest models for each distribution setting and apply `PC-tree` algorithm equipped with the CI tests $\psi^{Ber}$, $\psi^{Gauss}$, and $\psi^{NP}$ specified in Sections 4-5 to estimate the graph structure.

In Figure 1, we present the structure Hamming distance (SHD) between the estimated graph and the ground truth against sample size $n$ for various settings of number of nodes $d = \{20, 40, 60, 80, 100\}$. Each setting is averaged over $N = 50$ replications. The results show as the sample size increases, SHD consistently decreases for graphs with 20 to

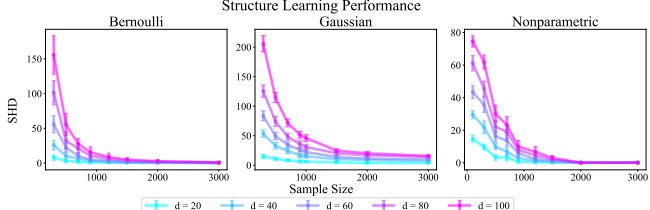

*Figure 1.* Structure Hamming distance (SHD) vs. sample size for poly-forest learning for Bernoulli, Gaussian, and nonparametric continuous distributions over varying number of nodes (indicated by colors). Error bars represent standard deviations. SHD consistently decreases toward zero as sample size increases across all experimental settings.

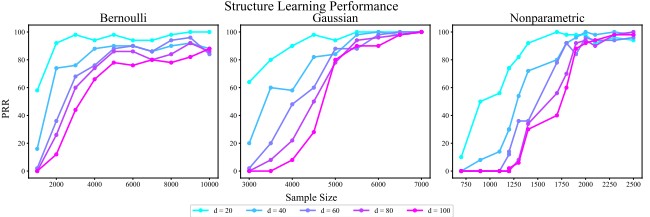

*Figure 2.* Precise Recovery Rate (PRR) vs. sample size for poly-forest learning for Bernoulli, Gaussian, and nonparametric continuous distributions over varying number of nodes (indicated by colors). PRR consistently increase toward 100% as sample size increases across all experimental settings.

100 nodes. A lower SHD means the output graph is closer to the truth, thus more accurate structure learning. The effect of dimensionality is illustrated by lines with different colors. Overall, our simulations demonstrate that combining a powerful conditional independence test $\psi$ with `PC-tree` algorithm allows for consistent and accurate poly-forest learning.

In addition to the SHD, we evaluate the Precise Recovery Rate (PRR), defined as the proportion of replications in which the estimated graph exactly captures the true Markov equivalence class. PRR thus provides a stringent notion of success, complementing SHD by measuring exact structure recovery rather than partial success. As shown in Figure 2, PRR consistently increases toward 100% as the sample size grows across all three distributional settings, Bernoulli, Gaussian, and nonparametric continuous, indicating convergence of the `PC-tree` algorithm to the correct structure.

As expected, larger graphs require more samples to attain high PRR; nevertheless, near-perfect recovery is achieved uniformly across all dimensions as $n$ increases. The convergence behavior differs across distributional regimes. While parametric models (Bernoulli and Gaussian) achieve high PRR with relatively fewer samples, the nonparametric model requires substantially more samples to reach comparable recovery rates, reflecting the intrinsic difficulty induced from nonparametric conditional independence testing

compared to the parametric ones predicted by our theory.

Additional experiments, details on implementations and how the data are simulated can be found in Appendix I.

## 7. Conclusion

In this paper, we study the minimax optimality of structure learning and conditional independence testing. We make their intuitive connection rigorous and quantitative by showing the optimality conditions for CI testing translate directly into the optimality of poly-forest learning, which can be achieved by an efficient constraint-based algorithm with the optimal CI test as input. The generic theoretical results are demonstrated using three representative distribution families. This finding highlights the central role of CI testing in structure learning in the view of statistical sample efficiency.

One interesting direction for future research is to extend the results beyond poly-forests to general DAGs. In such settings, constraint-based methods, such as the PC algorithm, are applicable and rely solely on CI testing to operate. We anticipate that a similar statistical connection between CI testing and structure learning can be established for general DAGs, though additional structural parameters, such as the maximum in-degree, are likely to play a key role in characterizing the corresponding optimal sample complexity.

## Acknowledgements

B.A. gratefully acknowledges NSF support through DMS-2610618, IIS-2453378, IIS-1956330, and the Robert H. Topel Faculty Research Fund.

## Impact Statement

This paper presents work whose goal is to advance the field of Machine Learning. There are many potential societal consequences of our work, none which we feel must be specifically highlighted here

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

---

**Algorithm 1** `PC-Tree` algorithm

---

**Input:** $n$ i.i.d. samples $\{X_1^{(i)}, \ldots, X_d^{(i)}\}_{i=1}^n$, CI test $\psi$ as function of data;

1. Let $\widehat{E} = \emptyset$.

2. For each pair $(j, k)$, $0 \le j < k \le d$:

   (a) For all $\ell \in [d] \cup \{\emptyset\} \setminus \{j, k\}$:
       i. Test $H_0 : X_j \perp\!\!\!\perp X_k \,|\, X_\ell$ vs. $H_1 : X_j \,\not\!\perp\!\!\!\perp\, X_k \,|\, X_\ell$ using $\psi$, store the results.
   (b) If all tests reject, then $\widehat{E} \leftarrow \widehat{E} \cup \{j - k\}$.
   (c) Else (if some test accepts), let $S(j, k) = \{\ell \in [d] \cup \{\emptyset\} \setminus \{j, k\} : X_j \perp\!\!\!\perp X_k \,|\, X_\ell\}$.

**Output:** $\widehat{G} = ([d], \widehat{E})$, separation set $S$.

---

---

**Algorithm 2** ORIENT algorithm

---

**Input:** Skeleton $\widehat{G}$, separation sets $S$
**Output:** CPDAG $\widehat{\widehat{G}}$.

1. For all pairs of nonadjacent nodes $j, k$ with common neighbour $\ell$:

   (a) If $\ell \notin S(j, k)$, then orient $j - \ell - k$ in $\widehat{G}$ by $j \to \ell \leftarrow k$

2. In the resulting PDAG $\widehat{G}$, orient as many as possible undirected edges by applying following rules:

   - **R1** Orient $k - \ell$ into $k \to \ell$ whenever there is an arrow $j \to k$ such that $j$ and $\ell$ are not adjacent
   - **R2** Orient $j - k$ into $j \to k$ whenever there is a chain $j \to \ell \to k$
   - **R3** Orient $j - k$ into $j \to k$ whenever there are two chains $j - \ell \to k$ and $j - i \to k$ such that $\ell$ and $i$ are not adjacent
   - **R4** Orient $j - k$ into $j \to k$ whenever there are two chains $j - \ell \to i$ and $\ell - i \to k$ such that $\ell$ and $i$ are not adjacent

3. Return $\widehat{G}$ as $\widehat{\widehat{G}}$.

---

## A. Details of `PC-tree` algorithm

Introduced in Wang et al. (2024), `PC-tree` algorithm (Algorithm 1) is a modification to the classical PC algorithm and tailored for learning poly-forests/trees. Generically, it takes observational data along with a valid CI test as input, and outputs the skeleton with a separation set. The estimated skeleton will be further oriented by rules specified in Algorithm 2 using the separation set.

`PC-tree` conducts CI tests to determine the presence of edge between any two nodes. To achieve this, it only tests marginal independence and conditional independence given only one other node, rather than considering all possible conditioning sets as in the classical PC algorithm. Therefore, `PC-tree` only invokes approximately

$$\binom{d}{2} \times \left[1 + (d - 2)\right] \asymp d^3$$

times of CI test $\psi$, thereby is efficient. Since poly-forests are effectively concatenation of poly-trees, `PC-tree` is also consistent for learning poly-forest structures. As long as the input CI test $\psi$ is valid and well controls the type-I and type-II errors, the consistency of the algorithm applies to general distributions.

## B. Details of applications

In this appendix, we detail the formal definitions of the model class considered in each application (Sections 4-5). Specifically, we specify $\mathcal{P}$ and $m$, thereby $\mathcal{H}_1$ and $\mathcal{H}_0$ will follow, for the exampled distributions.

## B.1. Bernoulli distribution (Section 4.1)

We start by specifying the model class $\mathcal{P}$. Consider all multivariate Bernoulli distributions with dimension being three. They are parametrized by joint probability mass function $p(x, y, z)$:

$$
\mathcal{P}^{Ber} = \Big\{ p(x, y, z) :
$$
$$
(X, Y, Z) \sim \mathbb{P}(X = x, Y = y, Z = z) = p(x, y, z),
$$
$$
(x, y, z) \in \{0, 1\}^3, \sum_{x, y, z} p(x, y, z) = 1 \Big\}.
$$

We measure dependence using the total variation distance to the product of its marginals:

$$
m^{Ber}(X, Y \mid Z) = \sum_z p(z) \sum_{x, y} |p(x, y \mid z) - p(x \mid z) p(y \mid z)|.
$$

Then $\mathcal{H}_0^{Ber}$ and $\mathcal{H}_1^{Ber}$ contain all these Bernoulli distributions that are conditionally independent and dependent respectively:

$$
\mathcal{H}_0^{Ber} = \{p \in \mathcal{P}^{Ber} : X \perp\!\!\!\perp Y \mid Z\}, \quad \mathcal{H}_1^{Ber} = \{p \in \mathcal{P}^{Ber} : m^{Ber}(X, Y \mid Z) \geq r\}
$$

Having defined the elements of CI testing problem $\mathcal{C}^{Ber} := \mathcal{C}(\mathcal{P}^{Ber}, m^{Ber}, n)$, the Bernoulli poly-forest learning problem $\mathcal{F}^{Ber} := \mathcal{F}(\mathcal{P}^{Ber}, m^{Ber}, c)$ contains all multivariate Bernoulli distributions with dimension $d$ and Markov and $c$-strong tree-faithful to some poly-forest $G \in \mathcal{T}$:

$$
(X_1, X_2 \ldots, X_d) \sim \mathbb{P}(X_1 = x_1, X_2 = x_2, \ldots, X_d = x_d), \quad (x_1, x_2 \ldots, x_d) \in \{0, 1\}^d.
$$

## B.2. Gaussian distribution (Section 4.2)

Consider all multivariate Gaussian distributions with dimension being three:

$$
\mathcal{P}^{Gauss} = \Big\{ p(X, Y, Z) : (X, Y, Z) \sim \mathcal{N}(\mathbf{0}_3, \Sigma), \Sigma \in \mathbb{S}_{++}^3 \Big\}.
$$

We measure dependence using the partial correlation:

$$
m^{Gauss}(X, Y \mid Z) = \frac{|\operatorname{cov}(X, Y \mid Z)|}{\sqrt{\operatorname{var}(X \mid Z) \operatorname{var}(Y \mid Z)}}. \tag{6}
$$

Then $\mathcal{H}_0^{Gauss}$ and $\mathcal{H}_1^{Gauss}$ contain all these Gaussian distributions that are conditionally independent and dependent respectively.

$$
\mathcal{H}_0^{Gauss} = \{p \in \mathcal{P}^{Gauss} : X \perp\!\!\!\perp Y \mid Z\}, \quad \mathcal{H}_1^{Gauss} = \{p \in \mathcal{P}^{Gauss} : m^{Gauss}(X, Y \mid Z) \geq r\}
$$

Consequently, the Gaussian CI testing problem $\mathcal{C}^{Gauss} := \mathcal{C}(\mathcal{P}^{Gauss}, m^{Gauss}, n)$ is defined. Meanwhile, it gives the Gaussian poly-forest learning problem $\mathcal{F}^{Gauss} := \mathcal{F}(\mathcal{P}^{Gauss}, m^{Gauss}, c)$, which include all multivariate Gaussian distributions with dimension $d$ and Markov and $c$-strong tree-faithful to some poly-forest $G \in \mathcal{T}$:

$$
(X_1, X_2 \ldots, X_d) \sim \mathcal{N}(\mathbf{0}_d, \Sigma), \Sigma \in \mathbb{S}_{++}^d.
$$

Similar to $\psi^{Ber}$, we construct a test by thresholding the sample partial correlation:

$$
\widehat{m}^{Gauss}(X, Y \mid Z) = \frac{|\widehat{\Sigma}_{XY} - \widehat{\Sigma}_{XZ}\widehat{\Sigma}_{ZZ}^{-1}\widehat{\Sigma}_{ZY}|}{\sqrt{(\widehat{\Sigma}_{XX} - \widehat{\Sigma}_{XZ}\widehat{\Sigma}_{ZZ}^{-1}\widehat{\Sigma}_{ZX})(\widehat{\Sigma}_{YY} - \widehat{\Sigma}_{YZ}\widehat{\Sigma}_{ZZ}^{-1}\widehat{\Sigma}_{ZY})}}
$$
$$
\psi^{Gauss} = \mathbb{1}\{\widehat{m}^{Gauss}(X, y \mid Z) \geq c/2\}, \tag{7}
$$

where $\widehat{\Sigma} = \frac{1}{n} \sum_{i=1}^n X^{(i)} X^{(i)^\top}$ is the sample covariance matrix.

### B.3. Nonparametric continuous distribution (Section 5)

We follow the CI testing framework in Neykov et al. (2021). Consider all continuous distributions over $[0,1]^3$ that admit continuous densities $p(X,Y,Z)$. In addition, we impose following two smoothness condition on the densities.

**Definition B.1** (Lipschitzness). For some constant $L_1 > 0$,

- if $X \perp\!\!\!\perp Y \mid Z$, then $\|p_{X \mid Z=z} - p_{X \mid Z=z'}\|_1 \leq L_1 |z - z'|$ and $\|p_{Y \mid Z=z} - p_{Y \mid Z=z'}\|_1 \leq L_1 |z - z'|$.

- if $X \not\perp\!\!\!\perp Y \mid Z$, then $\|p_{X,Y \mid Z=z} - p_{X,Y \mid Z=z'}\|_1 \leq L_1 |z - z'|$.

**Definition B.2** (*s*-Hölder smoothness). For some $s > 0$, let $\lfloor s \rfloor$ be the maximum integer smaller than $s$. For some constant $L_2 > 0$ and all $x, x', y, y', z \in [0,1]$,

$$\sup_{t \leq \lfloor s \rfloor} \left| \frac{\partial^t}{\partial x^t} \frac{\partial^{\lfloor s \rfloor - t}}{\partial y^{\lfloor s \rfloor - t}} p(x, y \mid z) - \frac{\partial^t}{\partial x^t} \frac{\partial^{\lfloor s \rfloor - t}}{\partial y^{\lfloor s \rfloor - t}} p(x', y' \mid z) \right| \leq L_2 \sqrt{(x - x')^2 + (y - y')^2}^{\,s - \lfloor s \rfloor}$$

and

$$\sup_{t \leq \lfloor s \rfloor} \left| \frac{\partial^t}{\partial x^t} \frac{\partial^{\lfloor s \rfloor - t}}{\partial y^{\lfloor s \rfloor - t}} p(x, y \mid z) \right| \leq L_2 \,.$$

The Lipschitzness is used to characterize the smoothness with respect to the conditioning variable $Z$, while the Hölder smoothness is required for the conditional densities with the conditioning variable fixed, and particularly for the (conditionally) dependent variable pairs. Denote this class of distributions as $\mathcal{P}^{NP}$. As specified in the main paper, we measure the dependence using

$$m^{NP}(X, Y \mid Z) = \inf_{q \in \mathcal{Q}} \|p - q\|_1$$

which is the distance to the closest (conditionally) independent distributions, and should be large enough such that the dependent distributions can be distinguished from the independent ones (Shah & Peters, 2020). Therefore, the CI testing class is defined $\mathcal{C}^{NP} := \mathcal{C}(\mathcal{P}^{NP}, m^{NP}, n)$, with the null and alternative hypotheses being

$$\mathcal{H}_0^{NP} = \{p \in \mathcal{P}^{NP} : X \perp\!\!\!\perp Y \mid Z \text{ and satisfies Lipschitzness}\}$$
$$\mathcal{H}_1^{NP} = \{p \in \mathcal{P}^{NP} : m^{NP}(X, Y \mid Z) \geq r \text{ and satisfies Lipschitzness and } s\text{-Hölder smoothness}\} \,.$$

Correspondingly, the nonparametric poly-forest learning problem is defined, which includes all continuous distributions supported on $[0,1]^d$ and Markov and $c$-strong tree-faithful to some poly-forest $G \in \mathcal{T}$, denoted as $\mathcal{F}^{NP} := \mathcal{F}(\mathcal{P}^{NP}, m^{NP}, c)$.

## C. Proof of Theorem 3.1

We provide the full version of Theorem 3.1 below, followed by the discussion and proof.

**Theorem C.1.** *Given a conditional independence testing problem $\mathcal{C}(\mathcal{P}, m, n)$, if for some constants $A$ and $A'$ independent with sample size $n$,*

*(C1) There exists a test $\psi$ such that when $c \geq A \times n^{-1/\alpha}$, it holds that*

$$\sup_{p \in \mathcal{H}_0} \mathbb{E}_p[\psi] + \sup_{p \in \mathcal{H}_1} \mathbb{E}_p[1 - \psi] \leq \frac{1}{2} \,;$$

*(C2) There exist $p_0 \in \mathcal{H}_0$ and $p_1 \in \mathcal{H}_1$ such that $\mathbf{KL}(p_1 \| p_0) \leq A' \times c^\alpha$;*

*then the minimax testing radius of $\mathcal{C}(\mathcal{P}, m, n)$ is $r_n \asymp n^{-1/\alpha}$, and $\psi$ is optimal for $\mathcal{C}(\mathcal{P}, m, n)$. In addition, if*

*(C3) $p_0$ and $p_1$ are Markov and $c$-strong tree-faithful to some poly-forests of three nodes;*

*then the optimal sample complexity of learning $\mathcal{F}(\mathcal{P}, m, c)$ is*

$$n \asymp \frac{\log d}{c^\alpha},$$

*which is achieved by* `PC-tree` *with $\psi$.*

The first two conditions assumed in Theorem C.1 are standard steps to study minimax optimality, corresponding to upper and lower bounds of testing radius. It is typical in literature to verify these two conditions when deriving the optimal testing radius:

- (C1) requires finding a powerful test to distinguish independent and dependent instances with sufficiently large signal.

- (C2) looks for two instances from $\mathcal{H}_0$ and $\mathcal{H}_1$ that are hard to distinguished. Typically, $p_0$ is a "flat" distribution, e.g. uniform distribution or Bern(0.5), while $p_1$ is constructed to be a perturbation to the flat distribution with sufficiently large signal $c$ but small deviation to being independent.

- With above two conditions, to conclude optimal poly-forest learning, (C3) requires the instances to be generated by poly-forest models, ensuring their validity in the context of structure learning.

*Proof.* Under the given conditions, we prove the optimalities for CI testing and poly-forest learning.

**Optimal CI testing**
In the context of CI testing, (C2) reads as $\mathbf{KL}(p_1 \| p_0) \leq A' \times r^\alpha$ (with a little abuse of notation to replace $c$ by $r$).

The existence of test $\psi$ implies a upper bound of testing radius $r_n \lesssim n^{-1/\alpha}$. The lower bound is given by Le Cam's two point method: by the existence $p_0 \in \mathcal{H}_0, p_1 \in \mathcal{H}_1$, we have

$$\inf_\psi \left\{ \sup_{p \in \mathcal{H}_0} \mathbb{E}_p[\psi] + \sup_{p \in \mathcal{H}_1} \mathbb{E}_p[1 - \psi] \right\} \geq \frac{1}{2} \Big( 1 - n\mathbf{TV}(p_1 \| p_0) \Big),$$

and $\mathbf{TV}(p_1 \| p_0) \leq \mathbf{KL}(p_1 \| p_0) \leq A' \times r^\alpha$. This requires the testing radius $n r_n^\alpha \gtrsim 1$, which yields $r_n \gtrsim n^{-1/\alpha}$.

**Optimal poly-forest learning**
*Regarding the upper bound*, we start with applying median trick (Motwani, 1995) to obtain an exponential decay of error probability. The idea is to divide the full sample into $K$ folds and apply the statistical test on each sub-sample. Take the majority vote of all these tests as the final output. Then the final output makes mistake only when half of the tests make mistake, whose probability can easily computed and bounded in an exponential way and serves our goal when $K$ is appropriately chosen.

**Proposition C.2.** *Suppose there exists a statistical test $\psi$ such that if the sample size $n \gtrsim N$, then*

$$\mathbb{P}(\psi \text{ is incorrect}) \leq 1/2,$$

*then there exists a CI test $\psi'$ such that under the same condition,*

$$\mathbb{P}(\psi' \text{ is incorrect}) \leq \exp\Big( - C_0 n / N \Big),$$

*for some constant $C_0$ independent with $n$.*

Applying Proposition C.2, we can derive an error probability bound for $\psi$ with modification. For simplicity, we will stick with $\psi$ with a little abuse of notation. Assuming $n \gtrsim c^{-\alpha}$, we have for some constant $C_0$,

$$\mathbb{P}(\psi \text{ is incorrect}) \leq \exp\Big( - C_0 n c^\alpha \Big).$$

With this error bound in hand, the proof proceeds by following the proof of Theorem 4.3 in Wang et al. (2024) with minor modifications: 1) replacing poly-tree with poly-forest; 2) replacing correlation CI testing with $\psi$. This completes the proof of upper bound, and shows `PC-tree` with $\psi$ as CI tester achieves the upper bound.

*Regarding the lower bound*, without loss of generality, we assume $d$ divides 3, otherwise proceed by setting the remaining one or two nodes to be isolated and follow $p_0(X)$. Group all the nodes into $d' = d/3$ clusters:

$$\{1, 2, 3\}, \{4, 5, 6\}, \cdots, \{3k+1, 3k+2, 3k+3\}, \cdots, \{3d'+1, 3d'+2, 3d'+3\}.$$

We are going to use the $p_1, p_0$ to parametrize the construction. Since $p_1$ and $p_0$ give different conditional independence statements, they are Markov and $c$-strong tree-faithful to two different poly-forests, which are denoted as $T_0, T_1$.

Firstly, construct graph $G_0$ by stacking $d'$ many $T_0$'s together. Then for each $k \geq 1$, construct graph $G_k$ by stacking $d' - 1$ many $T_0$'s and one $T_1$ together, and the subgraph $T_1$ is imposed on nodes $\{3k+1, 3k+2, 3k+3\}$, while $T_0$ is imposed on all remaining triplets. Under this construction, $\{G_k\}_{k \geq 0}$ are poly-forests with (at least) $d'$ many disconnected subgraphs, and are distinct with each other.

Now we consider distributions for each $k \geq 0$. Let $P_0 = p_0^{\otimes d'}$ and

$$P_k(X) = p_0^{\otimes d'-1} \times P_k(X_{3k+1}, X_{3k+2}, X_{3k+3}) \quad \text{with} \quad P_k(X_{3k+1}, X_{3k+2}, X_{3k+3}) = p_1.$$

Since $p_0$ and $p_1$ are Markov and $c$-strong tree-faithful to $T_0$ and $T_1$ respectively, $P_k$ is Markov and $c$-strong tree-faithful to $G_k$ for all $k \geq 0$. Therefore, $\{P_k\}_{k \geq 0} \in \mathcal{F}(\mathcal{P}, m, c)$.

We also going to apply Tsybakov's method (Corollary H.4). Firstly, we have the size of the construction to be lower bounded

$$\log M = \log(d' + 1) \geq \frac{1}{2} \log d.$$

Then we upper bound the KL divergence between $P_k$ and $P_0$ for all $k \geq 1$:

$$\mathbf{KL}(P_k \| P_0) = \mathbb{E}_{P_k} \log \frac{P_k}{P_0} = \mathbb{E}_{P_k} \log \frac{p_1}{p_0} = \mathbf{KL}(p_1 \| p_0) \leq A' \times c^\alpha.$$

Invoking Corollary H.4 completes the proof. $\qquad\square$

*Proof of Proposition C.2.* Divide the full sample into $K$ folds, which we will specify later, and apply $\psi_0$ for each of the sub-sample to get outputs $\psi^{(1)}, \ldots, \psi^{(K)}$. Let

$$\psi = \arg\max_{t \in \{0,1\}} \sum_{k=1}^{K} \mathbb{1}\{\psi^{(k)} = t\}$$

be the majority vote of all the tests as final output. Therefore, the incorrectness of $\psi$ implies at least half of the sub-tests make mistakes. Suppose that $n/K \gtrsim N$, by the guarantee of $\psi_0$, we have

$$\mathbb{P}(\psi \text{ is incorrect}) \leq \mathbb{P}\left(\text{half of } \{\psi^{(k)}\}_{k=1}^{K} \text{ is incorrect}\right)$$
$$\leq \frac{1}{2^{K/2}} = \exp\left(-\frac{\log 2}{2} \times K\right).$$

Now set $K = C'n/N$ for constant $C'$ large enough such that $n/K \gtrsim N$ is satisfied, let $C_0 = C'(\log 2)/2$, we obtain

$$\mathbb{P}(\psi \text{ is incorrect}) \leq \exp\left(-C_0 n/N\right)$$

as desired. $\qquad\square$

## D. Proof of Theorem 4.1 (Bernoulli distribution)

We start by showing the upper bound (C1) of $\psi^{Ber}$, then proceed to verify the validity of $p_0^{Ber}$ and $p_1^{Ber}$ and show they satisfy (C2) and (C3).

## D.1. Proof of upper bound for Bernoulli distribution

*Proof.* We directly show $\psi^{Ber}$ satisfies an exponential bound. Notice that by construction, the concentration of $\widehat{m}$ implies the correctness of testing:

$$|\widehat{m}^{Ber} - m^{Ber}| \le c/2 \implies \psi^{Ber} \text{ is correct.}$$

We aim to show $|\widehat{m}^{Ber} - m^{Ber}| \le c/2$ holds with high probability. To achieve this, we employee the result below:

**Lemma D.1** ([Devroye](https://doi.org/) (1983), Lemma 3). *Let* $(X_1, X_2, \ldots, X_k)$ *be a multinomial* $(n, p_1, p_2, \ldots, p_k)$ *random vector. Let* $\widehat{p} = (X_1, X_2, \ldots, X_k)/n$ *For all* $\epsilon \in (0, 1)$ *and all* $k$ *satisfying* $k/n \le \epsilon^2/20$, *we have*

$$\mathbb{P}(\|\widehat{p} - p\|_1 > \epsilon) \le 3\exp(-n\epsilon^2/25).$$

Applied to our setup, we consider the concentration for the joint distribution $p_{XYZ}$, which can be viewed as a multinomial distribution with dimension $k = 2^3 = 8$. Therefore, for $\epsilon$ such that $n \ge 160/\epsilon^2$, with probability at least $1 - 3\exp(-n\epsilon^2/25)$, we have

$$\|p_{XYZ} - \widehat{p}_{XYZ}\|_1 := \sum_{xyz} |p(x, y, z) - \widehat{p}(x, y, z)| \le \epsilon,$$

which also implies the concentration of $p_{WZ}$ ($W \in \{X, Y\}$) and $p_Z$. To see this, take $W = X$ for example:

$$\|\widehat{p}_{XZ} - p_{XZ}\|_1 = \sum_{xz} |\widehat{p}(x, z) - p(x, z)| = \sum_{xz} \left| \sum_y \left( \widehat{p}(x, y, z) - p(x, y, z) \right) \right|$$

$$\le \sum_{xz} \sum_y |\widehat{p}(x, w, z) - p(x, w, z)| = \|p_{XYZ} - \widehat{p}_{XYZ}\| \le \epsilon.$$

Similarly,

$$\|\widehat{p}_Z - p_Z\|_1 = \sum_z |\widehat{p}(z) - p(z)| = \sum_z \left| \sum_{xy} \left( \widehat{p}(x, y, z) - p(x, y, z) \right) \right|$$

$$\le \sum_z \sum_{xy} |\widehat{p}(x, w, z) - p(x, w, z)| = \|p_{XYZ} - \widehat{p}_{XYZ}\| \le \epsilon.$$

Since the estimator $\widehat{m}^{Ber}$ involves the estimation of $p_{XY \mid Z}$ and $p_{W \mid Z}$ for $W = X$ or $Y$, we proceed to bound the error of them. For any $(x, y, z) \in \{0, 1\}^3$,

$$
\begin{aligned}
|\widehat{p}(x, y \mid z) - p(x, y \mid z)| &= \left| \frac{\widehat{p}(x, y, z)}{\widehat{p}(z)} - \frac{p(x, y, z)}{p(z)} \right| \\
&= \left| \frac{\widehat{p}(x, y, z)}{\widehat{p}(z)} - \frac{\widehat{p}(x, y, z)}{p(z)} + \frac{\widehat{p}(x, y, z)}{p(z)} - \frac{p(x, y, z)}{p(z)} \right| \\
&\le \widehat{p}(x, y, z) \left| \frac{1}{\widehat{p}(z)} - \frac{1}{p(z)} \right| + \frac{1}{p(z)} |\widehat{p}(x, y, z) - p(x, y, z)| \\
&\le \widehat{p}(x, y, z) \frac{|\widehat{p}(z) - p(z)|}{p(z)\widehat{p}(z)} + \frac{1}{p(z)} |\widehat{p}(x, y, z) - p(x, y, z)| \\
&= \frac{1}{p(z)} \left( \widehat{p}(x, y \mid z)|\widehat{p}(z) - p(z)| + |\widehat{p}(x, y, z) - p(x, y, z)| \right) \\
&\le \frac{1}{p(z)} \left( |\widehat{p}(z) - p(z)| + |\widehat{p}(x, y, z) - p(x, y, z)| \right).
\end{aligned}
$$

Analogously, for $W = X$ or $Y$ and any $(w, z) \in \{0, 1\}^2$

$$
\begin{aligned}
|\widehat{p}(w \,|\, z) - p(w \,|\, z)| &= \left| \frac{\widehat{p}(w, z)}{\widehat{p}(z)} - \frac{p(w, z)}{p(z)} \right| \\
&= \left| \frac{\widehat{p}(w, z)}{\widehat{p}(z)} - \frac{\widehat{p}(w, z)}{p(z)} + \frac{\widehat{p}(w, z)}{p(z)} - \frac{p(w, z)}{p(z)} \right| \\
&\leq \widehat{p}(w, z) \left| \frac{1}{\widehat{p}(z)} - \frac{1}{p(z)} \right| + \frac{1}{p(z)} |\widehat{p}(w, z) - p(w, z)| \\
&\leq \widehat{p}(w, z) \frac{|\widehat{p}(z) - p(z)|}{p(z)\widehat{p}(z)} + \frac{1}{p(z)} |\widehat{p}(w, z) - p(w, z)| \\
&= \frac{1}{p(z)} \Big( \widehat{p}(w \,|\, z)|\widehat{p}(z) - p(z)| + |\widehat{p}(w, z) - p(w, z)| \Big) \\
&\leq \frac{1}{p(z)} \Big( |\widehat{p}(z) - p(z)| + |\widehat{p}(w, z) - p(w, z)| \Big).
\end{aligned}
$$

Denote $\widehat{p}(x, y, z) = p(x, y, z) + \delta_{xyz}$, $\widehat{p}(w, z) = p(w, z) + \delta_{wz}$, $\widehat{p}(z) = p(z) + \delta_z$, thus $\delta_{xyz}, \delta_{wz}$ are bounded correspondingly. Then we are ready to show the concentration of $\widehat{m}^{Ber}$ below.

$$
|\widehat{m}^{Ber} - m^{Ber}| = \sum_z \left\{ \widehat{p}(z) \sum_{xy} |\widehat{p}(x, y \,|\, z) - \widehat{p}(x \,|\, z)\widehat{p}(y \,|\, z)| - p(z) \sum_{xy} |p(x, y \,|\, z) - p(x \,|\, z)p(y \,|\, z)| \right\},
$$

where

$$
\begin{aligned}
&\widehat{p}(z) \sum_{xy} |\widehat{p}(x, y \,|\, z) - \widehat{p}(x \,|\, z)\widehat{p}(y \,|\, z)| \\
=& \Big( p(z) + \delta_z \Big) \sum_{xy} \left| \Big( p(x, y \,|\, z) + \delta_{xyz} \Big) - \Big( p(x \,|\, z) + \delta_{xz} \Big) \Big( p(y \,|\, z) + \delta_{yz} \Big) \right| \\
\leq& p(z) \sum_{xy} |p(x, y \,|\, z) - p(x \,|\, z)p(y \,|\, z)| + p(z) \sum_{xy} \left\{ |\delta_{xyz}| + |\delta_{xz}\delta_{yz} + \delta_{xz}p(y \,|\, z) + \delta_{yz}p(x \,|\, z)| \right\} \\
&+ \delta_z \sum_{xy} |\widehat{p}(x, y \,|\, z) - \widehat{p}(x \,|\, z)\widehat{p}(y \,|\, z)| \\
\leq& p(z) \sum_{xy} |p(x, y \,|\, z) - p(x \,|\, z)p(y \,|\, z)| + p(z) \sum_{xy} \left\{ |\delta_{xyz}| + |\delta_{xz}\widehat{p}(y \,|\, z) + \delta_{yz}p(x \,|\, z)| \right\} + 8\delta_z \\
\leq& p(z) \sum_{xy} |p(x, y \,|\, z) - p(x \,|\, z)p(y \,|\, z)| + p(z) \sum_{xy} \left\{ |\delta_{xyz}| + |\delta_{xz}| + |\delta_{yz}| \right\} + 8\delta_z.
\end{aligned}
$$

Therefore,

$$
\begin{aligned}
|\widehat{m}^{Ber} - m^{Ber}| &\leq \sum_z \left( p(z) \sum_{xy} \left\{ |\delta_{xyz}| + |\delta_{xz}| + |\delta_{yz}| \right\} + 8\delta_z \right) \\
&\leq 4\|p_Z - \widehat{p}_Z\|_1 + \|p_{XYZ} - \widehat{p}_{XYZ}\|_1 \\
&\quad 4\|p_Z - \widehat{p}_Z\|_1 + 2\|p_{XZ} - \widehat{p}_{XZ}\|_1 \\
&\quad 4\|p_Z - \widehat{p}_Z\|_1 + 2\|p_{YZ} - \widehat{p}_{YZ}\|_1 \\
&\quad 8\|p_Z - \widehat{p}_Z\|_1 \\
&= 20\|p_Z - \widehat{p}_Z\|_1 + \|p_{XYZ} - \widehat{p}_{XYZ}\|_1 + 2\|p_{XZ} - \widehat{p}_{XZ}\|_1 + +2\|p_{YZ} - \widehat{p}_{YZ}\|_1 \\
&\leq 25\epsilon.
\end{aligned}
$$

Choosing $\epsilon = c/50$, we have with probability at least

$$
1 - 3\exp(-C_0 nc^2),
$$

$|\widehat{m}^{Ber} - m^{Ber}| \leq c/2$ and $\psi^{Ber}$ is correct, where the constant $C_0 = 50^2 \times 25$. $\qquad\square$

### D.2. Proof of lower bound for Bernoulli distribution

*Proof.* We proceed to verify the validity of $p_0^{Ber}$ and $p_1^{Ber}$ by showing they satisfy (C2) and (C3). We can compute the KL divergence between them:

$$\mathbf{KL}(p_1^{Ber} \| p_0^{Ber}) = \log(1 - 4c^2) + 2c \log(1 + \frac{4c}{1 - 2c}) \leq 100c^2 \, ,$$

for $c$ small enough. Then (C2) holds. Moreover, it suffices to show for $p_1^{Ber}$ is $c$-strong tree-faithfulness to the chain graph $Z \to X \to Y$. To achieve this, we can compute

$$m^{Ber}(X, Y) = 2c, m^{Ber}(X, Z) = 2c,$$
$$m^{Ber}(X, Y \mid Z) \geq 2c(1 - 4c^2) \geq c,$$
$$m^{Ber}(Z, X \mid Y) \geq 2c(1 - 4c^2) \geq c,$$

for $c$ small enough. Then (C3) holds. $\qquad\square$

## E. Proof of Theorem 4.2 (Gaussian distribution)

*Proof.* The upper bound (C1) of $\psi^{Gauss}$ is given by Lemma C.1 in Wang et al. (2024). We proceed to verify the validity of $p_0^{Gauss}$ and $p_1^{Gauss}$ by showing they satisfy (C2) and (C3). We can compute the KL divergence between them:

$$\mathbf{KL}(p_1^{Gauss} \| p_0^{Gauss}) = -\log(1 - 4c^2) \leq 100c^2 \, ,$$

for $c$ small enough. Then (C2) holds. Moreover, it suffices to show for $p_1^{Gauss}$ is $c$-strong tree-faithfulness to the chain graph $Z \to X \to Y$. To achieve this, we can compute

$$m^{Gauss}(X, Y) = 2c, m^{Gauss}(X, Z) = 2c,$$
$$m^{Gauss}(X, Y \mid Z) \geq 2c(1 - 4c^2) \geq c,$$
$$m^{Gauss}(Z, X \mid Y) \geq 2c(1 - 4c^2) \geq c,$$

for $c$ small enough. Then (C3) holds. $\qquad\square$

## F. Proof of Theorem 4.3 (poly-tree learning)

*Proof.* The optimality of Gaussian poly-tree learning is established in Wang et al. (2024). For Bernoulli poly-tree learning, the upper bound using $\psi^B$ follows the proof of Theorem 4.3 in Wang et al. (2024) combined with results in Appendix D.1. It suffices to prove the validity of the lower bound construction (5), which is given by the following two lemmas:

**Lemma F.1.** *Assuming $c \leq 1/4$, for any two Markov chains $T_1, T_2$, $\mathbf{KL}(P_{T_1} \| P_{T_2}) \leq 16dc^2$.*

**Lemma F.2.** *Assuming $c \leq 1/4$, for any Markov chain $T$, $P_T$ satisfies $c$-strong tree-faithfulness with respect to $m^B$.*

Since the size of all directed Markov chains is

$$\log d! \geq \frac{1}{2} d \log d \, ,$$

Corollary H.3 implies the lower bound of $\log d / c^2$. $\qquad\square$

We proceed to prove Lemma F.1 and F.2. We start with three facts about the construction: the first states the marginal of any variable is a "coin flip"; the second states the positions of conditional probability can be switched; the third states the conditional probability of each variable is close to "coin flip".

**Lemma F.3.** *For $P_T$, we have the following:*

1. *For any $k \in [d]$, $\mathbb{P}(X_k = x_k) = 1/2$ for $x_k = 1$ or $0$.*

2. *For any $j \neq k \in [d]$, $\mathbb{P}(X_k = x_k \mid X_j = x_j) = \mathbb{P}(X_j = x_j \mid X_k = x_k)$.*

3. *For any $j \neq k \in [d]$, $\mathbb{P}(X_k = x_k \mid X_j = x_j) = 1/2 + \delta$ with $|\delta| \leq c$, for $x_k, x_j \in \{0, 1\}$.*

*Proof.* Without loss of generality, fix $T$ to be the natural ordering: $1 \to 2 \to \cdots \to d$. For the first fact, since $X_1$ is a Bernoulli random variable, for $X_2$,

$$\mathbb{P}(X_2 = x_2) = \sum_{x_1} \mathbb{P}(X_2 = x_2 \mid X_1 = x_1)\mathbb{P}(X_1 = x_1) = \frac{1}{2} \times (\frac{1}{2} + c) + \frac{1}{2} \times (\frac{1}{2} - c) = \frac{1}{2} .$$

So marginally $X_2$ is a Bernoulli random variable. By induction, $X_3, X_4, \ldots, X_d$ are all marginally a Bernoulli random variable.

For the second fact, because of the first fact,

$$\mathbb{P}(X_k \mid X_j) = \frac{\mathbb{P}(X_j \mid X_k)\mathbb{P}(X_k)}{\mathbb{P}(X_j)} = \mathbb{P}(X_j \mid X_k) .$$

For the last fact, due to the second fact, we look at the case where $j < k$, otherwise we can switch the positions. By Markov property,

$$\mathbb{P}(X_k = x_k \mid X_j = x_j) = \mathbb{P}(X_k = x_k \mid X_{k-1} = 1)\mathbb{P}(X_{k-1} = 1 \mid X_j = x_j)$$
$$+ \mathbb{P}(X_k = x_k \mid X_{k-1} = 0)\mathbb{P}(X_{k-1} = 0 \mid X_j = x_j) ,$$

which is a convex combination of $\frac{1}{2} \pm c$, then $\mathbb{P}(X_k = x_k \mid X_j = x_j) = \frac{1}{2} + \delta$ for some $|\delta| \leq c$. □

*Proof of Lemma F.1.* Without loss of generality, let $T_1$ be the natural ordering: $1 \to 2 \to \cdots \to d$, and $T_2$ is some other ordering $\pi$: $\pi(1) \to \pi(2) \to \cdots \to \pi(d)$. Then we have the KL divergence

$$\mathbf{KL}(P_{T_1} \| P_{T_2}) = \mathbb{E}_{P_{T_1}} \log \frac{P_{T_1}}{P_{T_2}} = \sum_{k=1}^{d} \left( \mathbb{E}_{T_1} \log 2P_{T_1}(X_k \mid X_{k-1}) - \mathbb{E}_{T_1} \log 2P_{T_2}(X_{\pi(k)} \mid X_{\pi(k-1)}) \right) .$$

Inside the summation, for the first term,

$$\mathbb{E}_{T_1} \log 2P_{T_1}(X_k \mid X_{k-1})$$
$$= \sum_{x_{k-1}} \mathbb{P}(X_{k-1} = x_{k-1}) \sum_{x_k} \mathbb{P}(X_k = x_k \mid X_{k-1} = x_{k-1}) \log 2\mathbb{P}(X_k = x_k \mid X_{k-1} = x_{k-1})$$
$$= 2 \times \frac{1}{2} \times \left( (\frac{1}{2} + c) \log(1 + 2c) + (\frac{1}{2} - c) \log(1 - 2c) \right)$$
$$= \frac{1}{2} \log(1 - 4c^2) + c \log(1 + \frac{4c}{1 - 2c}) .$$

For the second term, for simplicity, let $(\pi(k), \pi(k-1)) = (\ell, j)$. By Lemma F.3, suppose $\mathbb{P}(X_\ell = x_\ell \mid X_j = x_j) = 1/2 \pm \delta$. Then

$$\mathbb{E}_{T_1} \log 2P_{T_2}(X_{\pi(k)} \mid X_{\pi(k-1)})$$
$$= \sum_{x_j} \mathbb{P}(X_j = x_j) \sum_{x_\ell} \mathbb{P}(X_\ell = x_\ell \mid X_j = x_j) \log 2\mathbb{P}_{T_2}(X_\ell = x_\ell \mid X_j = x_j)$$
$$= 2 \times \frac{1}{2} \times \left( (\frac{1}{2} + \delta) \log(1 + 2c) + (\frac{1}{2} - \delta) \log(1 - 2c) \right)$$
$$= \frac{1}{2} \log(1 - 4c^2) + \delta \log(1 + \frac{4c}{1 - 2c}) .$$

Then the difference between the two terms is

$$(c - \delta) \log(1 + \frac{4c}{1 - 2c}) \leq (c - \delta)c \times \frac{4}{1 - 2c} \leq 16c^2 ,$$

when $c \leq 1/4$. Therefore, $\mathbf{KL}(P_{T_1} \| P_{T_2}) \leq 16dc^2$. □

*Proof of Lemma F.2.* Without loss of generality, let $T$ be the natural ordering: $1 \to 2 \to \cdots \to d$. Since there is no V-structure in Markov chain, we only need to show the first requirement in Definition 2.2 holds, i.e. for any $k$ and $j \neq k, k+1$, $m^B(X_k; X_{k+1} \mid X_j) \geq c$. There are two cases to look at: (1) $k > j$; (2) $j > k + 1$.

For the first case, we omit the capital letter when writing the probability.

$$
\begin{aligned}
m^B(X_k; X_{k+1} \mid X_j) &= \sum_{x_k, x_{k+1}, x_j} P(x_j) \Big| P(x_{k+1}, x_k \mid x_j) - P(x_{k+1} \mid x_j) P(x_k \mid x_j) \Big| \\
&= \sum_{x_j, x_k} P(x_j) P(x_k \mid x_j) \sum_{x_{k+1}} \Big| P(x_{k+1} \mid x_j, x_k) - P(x_{k+1} \mid x_j) \Big| \\
&= \sum_{x_j, x_k} P(x_j) P(x_k \mid x_j) \sum_{x_{k+1}} \Big| P(x_{k+1} \mid x_k) - \sum_{x_k'} P(x_{k+1} \mid x_k') P(x_k' \mid x_j) \Big| .
\end{aligned}
$$

By Lemma F.3, suppose $P(x_k \mid x_j) = 1/2 \pm \delta$. The calculation for the equation above is summarized below:

| $(x_j, x_k)$ | $P(x_j)P(x_k \mid x_j)$ | $\sum_{x_{k+1}} \Big| P(x_{k+1} \mid x_k) - \sum_{x_k'} P(x_{k+1} \mid x_k') P(x_k' \mid x_j) \Big|$ |
|---|---|---|
| $(1,1)$ | $\frac{1}{2}(\frac{1}{2} + \delta)$ | $2c(1 - 2\delta)$ |
| $(1,0)$ | $\frac{1}{2}(\frac{1}{2} - \delta)$ | $2c(1 + 2\delta)$ |
| $(0,1)$ | $\frac{1}{2}(\frac{1}{2} - \delta)$ | $2c(1 + 2\delta)$ |
| $(0,0)$ | $\frac{1}{2}(\frac{1}{2} + \delta)$ | $2c(1 - 2\delta)$ |

Therefore, we have

$$
m^B(X_k; X_{k+1} \mid X_j) = 2c(1 - 4\delta^2) \geq 2c(1 - 4c^2) \geq c,
$$

when $2(1 - 4c^2) \geq 1 \iff c^2 \leq 1/8$, which is implied by $c \leq 1/4$.

For the second case, we can rewrite

$$
\begin{aligned}
m^B(X_k; X_{k+1} \mid X_j) &= \sum_{x_k, x_{k+1}, x_j} \Big| P(x_{k+1}, x_k, x_j) - P(x_{k+1} \mid x_j) P(x_k, x_j) \Big| \\
&= \sum_{x_k, x_{k+1}, x_j} \Big| P(x_j \mid x_{k+1}) P(x_{k+1} \mid x_k) P(x_k) - P(x_j \mid x_k) P(x_k) P(x_{k+1} \mid x_j) \Big| \\
&= \sum_{x_k, x_{k+1}, x_j} P(x_k) P(x_j \mid x_{k+1}) \Big| P(x_{k+1} \mid x_k) - \sum_{x_{k+1}'} P(x_j \mid x_{k+1}') P(x_{k+1}' \mid x_k) \Big| .
\end{aligned}
$$

Again, by Lemma F.3, suppose $P(x_j \mid x_{k+1}) = 1/2 \pm \delta$. The calculation is summarized in the table below:

| $(x_k, x_{k+1}, x_j)$ | $P(x_j \mid x_{k+1})$ | $\Big| P(x_{k+1} \mid x_k) - \sum_{x_{k+1}'} P(x_j \mid x_{k+1}') P(x_{k+1}' \mid x_k) \Big|$ |
|---|---|---|
| $(1,1,1)$ | $\frac{1}{2} + \delta$ | $c(1 + 2\delta)$ |
| $(1,1,0)$ | $\frac{1}{2} - \delta$ | $c(1 - 2\delta)$ |
| $(1,0,1)$ | $\frac{1}{2} - \delta$ | $c(1 + 2\delta)$ |
| $(1,0,0)$ | $\frac{1}{2} + \delta$ | $c(1 - 2\delta)$ |
| $(0,1,1)$ | $\frac{1}{2} + \delta$ | $c(1 - 2\delta)$ |
| $(0,1,0)$ | $\frac{1}{2} - \delta$ | $c(1 + 2\delta)$ |
| $(0,0,1)$ | $\frac{1}{2} - \delta$ | $c(1 - 2\delta)$ |
| $(0,0,0)$ | $\frac{1}{2} + \delta$ | $c(1 + 2\delta)$ |

Therefore, we have

$$
m^B(X_k; X_{k+1} \mid X_j) = 4c > c,
$$

and $P_T$ satisfies $c$-strong tree-faithfulness. $\qquad \square$

# G. Proof of Theorem 5.1 (nonparametric continuous distribution)

*Proof.* The upper bound (C1) of $\psi^{NP}$ is given by Theorem 5.6 in Neykov et al. (2021). We proceed to specify the constructions of $p_0^{NP}$ and $p_1^{NP}$ and show they satisfy (C2) and (C3).

Let $p_0^{NP}$ be independent uniform distributions $Unif^3[0,1]$. Thus, $p_0^{NP}$ is Markov to an empty graph, and satisfies Lipschitz and smoothness condition. For $p_1^{NP}$, we design it to be Markov to a $V$-structure $X \to Z \leftarrow Y$, which is a three-node poly-forest. Under this graph, a faithful distribution is supposed to have $X \perp\!\!\!\perp Y$ while $X \not\perp\!\!\!\perp Y \mid Z$ and $X \not\perp\!\!\!\perp Z, Y \not\perp\!\!\!\perp Z$. In particular, $p_1^{NP}$ is specified as follows (we will suppress the superscript and subscript by writing it as $p$ to avoid notation clutter): $X, Y \sim Unif[0,1]$ and $p_{Z \mid X, Y}$ is a mixture of perturbation to uniform distribution, whose component depends on independent multi-dimensional Radmacher random variables $\Delta \in \{-1, 1\}^{m' \times m'}, \nu \in \{-1, 1\}^m$ for some positive integers $m, m'$ which are specified later. Then

$$p(Z \mid X, Y) = \mathbb{E}_{\Delta, \nu}\left[1 + \gamma_\Delta(X, Y)\eta_\nu(Z)\right],$$

where

$$\gamma_\Delta(x, y) = \rho^2 \sum_{i \in [m']} \sum_{j \in [m']} \Delta_{ij} h_{ij, m'}(x, y)$$

$$\eta_\nu(z) = \rho \sum_{j \in [m]} \nu_j h_{j, m}(z)$$

$$h_{ij, m'}(x, y) = \begin{cases} \sqrt{m'}^2 \widetilde{h}(m'x - i + 1, m'y - j + 1) & \forall (x, y) \in [\frac{i-1}{m}, \frac{i}{m}] \otimes [\frac{j-1}{m}, \frac{j}{m}] \\ 0 & \text{otherwise} \end{cases}$$

$$h_{j, m}(z) = \begin{cases} \sqrt{m} h(mz - j + 1) & \forall z \in [\frac{j-1}{m}, \frac{j}{m}] \\ 0 & \text{otherwise} \end{cases}$$

$$\int \widetilde{h}(x, y)dx = \sqrt{m'}h(y) \quad \int \widetilde{h}(x, y)dy = \sqrt{m'}h(x).$$

for some function $h(x)$ infinitely differentiable on $[0,1]$ such that

$$\int h(x)dx = 0, \int h^2(x)dx = 1, \int |h(x)|dx = b_1, \int |\widetilde{h}(x, y)|dxdy = b_2, \|h\|_\infty \vee \|h'\|_\infty \leq a,$$

for some $\rho > 0$ that will be specified later, and some constants $a, b_1, b_2 > 0$.

Now we show that each $p$ satisfy the $c$-strong tree-faithfulness and smoothness conditions. Since the operation $\mathbb{E}_{\Delta, \nu}$ is linear, we consider one instance of $(\Delta, \nu)$ in the following discussion.

$c$-**strong tree-faithfulness** We highlight several important observations:

$$p(z) = \int_{x, y} p(z \mid x, y)p(x)p(y) = \int_{x, y} p(z \mid x, y) = 1$$

$$p(x \mid z) = \frac{p(x, z)}{p(z)} = p(x, z) = \int_y p(z \mid x, y) = 1 + \left[\rho^2 \sum_i \left(\sum_j \Delta_{ij}\right) h_{i, m'}(x)\right]\eta_\nu(z)$$

$$p(y \mid z) = \frac{p(y, z)}{p(z)} = p(y, z) = \int_x p(z \mid x, y) = 1 + \left[\rho^2 \sum_j \left(\sum_i \Delta_{ij}\right) h_{j, m'}(y)\right]\eta_\nu(z).$$

Therefore, tree-faithfulness is satisfied (while strong version still needs to be shown):

$$p(x,z) - p(x)p(z) = \left[\rho^2 \sum_i \left(\sum_j \Delta_{ij}\right) h_{i,m'}(x)\right]\eta_\nu(z) \neq 0$$

$$p(y,z) - p(y)p(z) = \left[\rho^2 \sum_j \left(\sum_i \Delta_{ij}\right) h_{j,m'}(y)\right]\eta_\nu(z) \neq 0$$

$$p(x,y \mid z) - p(x \mid z)p(y \mid z) = \left\{\gamma_\Delta(x,y)\right.$$
$$\left. - \left[\rho^2 \sum_j \left(\sum_i \Delta_{ij}\right) h_{j,m'}(y)\right] - \left[\rho^2 \sum_i \left(\sum_j \Delta_{ij}\right) h_{i,m'}(x)\right]\right\}\eta_\nu(z)$$
$$- \rho^4 \left[\sum_i \left(\sum_j \Delta_{ij}\right) h_{i,m'}(x)\right]\left[\sum_j \left(\sum_i \Delta_{ij}\right) h_{j,m'}(y)\right]\eta_\nu^2(z) \neq 0 \,.$$

By Lemma B.4 in Neykov et al. (2021), it suffices to show the above three nonzero quantities are bounded away from zero in $L_1$. Specifically, because we have for any $i$ or $j$,

$$0.7\sqrt{m'} \leq \mathbb{E}_\Delta |\sum_j \Delta_{ij}| = \mathbb{E}_\Delta |\sum_i \Delta_{ij}| \leq \sqrt{m'} \,.$$

Then

$$\|p(x,y \mid z) - p(x \mid z)p(y \mid z)\|_1 \geq \rho^2 \int \left|\sum_{i,j} \Delta_{ij}\left[h_{ij,m'}(x,y) - h_{i,m'}(x) - h_{j,m'}(y)\right]\right|\left|\eta_v(z)\right|$$
$$- \rho^4 \int \left|\left[\sum_i \left(\sum_j \Delta_{ij}\right) h_{i,m'}(x)\right]\left[\sum_j \left(\sum_i \Delta_{ij}\right) h_{j,m'}(y)\right]\right|\left|\eta_v^2(z)\right|$$
$$\geq \left(\rho^2 (m')^2 \times \frac{1}{m'}g\right) \times \left(\rho\sqrt{m}a\right) - \rho^4 \left(m' \times \sqrt{m'} \times \frac{1}{\sqrt{m'}}a\right)^2 \times \left(\rho^2 m\right)$$
$$= \rho^3 m' \sqrt{m} \times ag - \rho^6 (m')^2 m \times a^2 \,,$$

where $g = \int |\widetilde{h}(x,y) - \frac{1}{\sqrt{m'}}h(x) - \frac{1}{\sqrt{m'}}h(y)|dxdy$ being a constant. And we need for either $w = x$ or $y$,

$$\|p(w,z) - p(w)p(z)\|_1 = \int \rho^2 \left|\sum_i \left(\sum_j \Delta_{ij}\right) h_{i,m'}(w)\right|\left|\eta_\nu(z)\right|$$
$$\geq \left(\rho^2 m' \times 0.7\sqrt{m'} \times \frac{1}{\sqrt{m'}}a\right) \times \left(\rho\sqrt{m}a\right)$$
$$= \rho^3 m' \sqrt{m}a^2 \,.$$

We will lower bound both of them above by the order of $c$ when specifying the parameters.

**Lipschitz condition**  We then check the TV distance between $p(x,y \mid z)$ and $p(x,y \mid z')$:

$$\|p(x,y \mid z) - p(x,y \mid z')\|_1 \leq \int \left|\gamma_\Delta(x,y)\right|\left|\eta_\nu(z) - \eta_\nu(z')\right|$$
$$\leq b_2 m' \rho^2 \times \left|\eta_\nu(z) - \eta_\nu(z')\right|$$
$$\leq \left[(b_2 m' \rho^2) \times (m^{1/2} m\rho\|h'\|_\infty)\right]|z - z'| \,.$$

We will need the term in the bracket to be smaller than some constant.

**Smoothness condition** We check the Hölder smoothness of $p(x, y \mid z)$ in $(x, y)$. Following the proof Theorem 4.2 in Neykov et al. (2021), for any $k \leq \lfloor s \rfloor$

$$\left| \frac{\partial^k}{\partial x^k} \frac{\partial^{\lfloor s \rfloor - k}}{\partial y^{\lfloor s \rfloor - k}} p(x, y \mid z) - \frac{\partial^k}{\partial x^k} \frac{\partial^{\lfloor s \rfloor - k}}{\partial y^{\lfloor s \rfloor - k}} p(x', y' \mid z) \right|$$

$$\leq \left| \frac{\partial^k}{\partial x^k} \frac{\partial^{\lfloor s \rfloor - k}}{\partial y^{\lfloor s \rfloor - k}} \gamma_\Delta(x, y)\eta_\nu(z) - \frac{\partial^k}{\partial x^k} \frac{\partial^{\lfloor s \rfloor - k}}{\partial y^{\lfloor s \rfloor - k}} \gamma_\Delta(x', y')\eta_\nu(z) \right|$$

$$\leq m^{1/2} \rho \|h\|_\infty \left| \frac{\partial^k}{\partial x^k} \frac{\partial^{\lfloor s \rfloor - k}}{\partial y^{\lfloor s \rfloor - k}} \gamma_\Delta(x, y) - \frac{\partial^k}{\partial x^k} \frac{\partial^{\lfloor s \rfloor - k}}{\partial y^{\lfloor s \rfloor - k}} \gamma_\Delta(x', y') \right|$$

$$\leq (\rho m^{1/2} a) \times \left( \rho^2 (m')^s \sqrt{m'}^2 \left\| \frac{\partial^k}{\partial x^k} \frac{\partial^{\lfloor s \rfloor - k}}{\partial y^{\lfloor s \rfloor - k}} \widetilde{h}(x, y) \right\|_\infty \right).$$

Thus, the quantity that need to be bounded above by constant is

$$\rho^3 m^{1/2} {m'}^{1+s} .$$

**Parameter choice** All in all, we need the choice of $m, m', \rho$ to satisfy the following requirements:

- $c$-strong tree-faithfulness:

$$\rho^3 m^{1/2} m' - \rho^6 m (m')^2 \gtrsim c$$
$$\rho^3 m^{1/2} m' \gtrsim c$$

- Lipschitz condition:

$$\rho^3 m^{3/2} m' \lesssim 1$$

- Smoothness condition:

$$\rho^3 m^{1/2} {m'}^{1+s} \lesssim 1$$

By setting $m' = m^{1/s}, \rho^3 \asymp m^{-(3/2 + 1/s)}, m \asymp c^{-1}$ and assuming $c$ is sufficiently small, above requirements are satisfied. Thus, $p$ falls into the considered model class.

**KL divergence** We start by bounding the $\chi^2$ divergence then upper bound KL divergence by $\chi^2$ divergence:

$$\chi^2(p_1^{NP} \| p_0^{NP}) + 1 = \int \frac{(p_1^{NP})^2}{p_0^{NP}} = \mathbb{E}_{\Delta, \Delta', \nu, \nu'} \int \left[ 1 + \gamma_\Delta(x, y)\eta_\nu(z) \right] \times \left[ 1 + \gamma_{\Delta'}(x, y)\eta_{\nu'}(z) \right].$$

The integral is

$$\int \left[ 1 + \gamma_\Delta(x, y)\gamma_{\Delta'}(x, y)\eta_\nu(z)\eta_{\nu'}(z) \right] = 1 + \rho^6 \langle \Delta, \Delta' \rangle \langle \nu, \nu' \rangle .$$

Therefore,

$$\chi^2(p_1^{NP} \| p_0^{NP}) + 1 = \mathbb{E}_{\Delta, \Delta', \nu, \nu'} \left\{ 1 + \rho^6 \langle \Delta, \Delta' \rangle \langle \nu, \nu' \rangle \right\} \leq \mathbb{E}_{\Delta, \Delta', \nu, \nu'} \left\{ \exp\left( \rho^6 \langle \Delta, \Delta' \rangle \langle \nu, \nu' \rangle \right) \right\} .$$

Following the proof Theorem 4.2 in Neykov et al. (2021), we can upper bound the right hand side above and obtain

$$\chi^2(p_1^{NP} \| p_0^{NP}) + 1 \leq \sqrt{\frac{1}{1 - (\rho^6)^2 m {m'}^2}} .$$

Since the function $f(t) = 1/\sqrt{1 - t^2} \leq 2t + 1$ for $t$ sufficiently small, with the choice of $\rho, m, m'$, we have for sufficiently small $c$,

$$\chi^2(p_1^{NP} \| p_0^{NP}) + 1 \leq C_0 \times c^{\frac{5s+2}{2s}} + 1\,,$$

for some constant $C_0$. Therefore, we arrive at

$$\mathbf{KL}(p_1^{NP} \| p_0^{NP}) \leq \chi^2(p_1^{NP} \| p_0^{NP}) \lesssim c^{\frac{5s+2}{2s}}$$

Application of Corollary H.4 completes the proof. $\qquad\square$

## H. Auxiliary lemmas

For lower bound techniques, we mainly apply the Fano's inequality and Tsybokov's method.

**Lemma H.1** (Yu (1997), Lemma 3). *For a model family $\mathcal{M}$ contains $M$ many distributions indexed by $j = 1, 2, \ldots, M$ such that*

$$\alpha = \max_{P_j \neq P_k \in \mathcal{M}} \mathbf{KL}(P_j \| P_k)$$
$$s = \min_{P_j \neq P_k \in \mathcal{M}} \mathbf{dist}(\theta(P_j), \theta(P_k))\,,$$

*where $\theta$ is a functional of its distribution argument. Then for any estimator $\widehat{\theta}$ for $\theta(P)$,*

$$\inf_{\widehat{\theta}} \sup_{P \in \mathcal{M}} \mathbb{E}_P \mathbf{dist}(\theta(P), \widehat{\theta}) \geq \frac{s}{2}\left(1 - \frac{\alpha + \log 2}{\log M}\right).$$

**Lemma H.2** (Tsybakov (2008), Theorem 2.5). *For a model family $\mathcal{M}$ contains distributions $P_0, P_1, \ldots, P_M$ with $M \geq 2$ and suppose that $\Theta$ contains elements $\theta_0, \theta_1, \ldots, \theta_M$ such that:*

1. $\mathbf{dist}(\theta_j, \theta_k) \geq 2s, \forall 0 \leq j < k \leq M$;

2. $P_j \ll P_0, \forall j = 1, \ldots, M$, *and*

$$\frac{1}{M} \sum_{j=1}^{M} \mathbf{KL}(P_j, P_0) \leq \alpha \log M$$

*with $0 < \alpha < 1/8$ and $P_j = P_{\theta_j}$. Then*

$$\inf_{\widehat{\theta}} \sup_{\theta \in \Theta} \mathbb{P}_\theta\left(\mathbf{dist}(\theta, \widehat{\theta}) \geq s\right) \geq \frac{\sqrt{M}}{1 + \sqrt{M}}\left(1 - 2\alpha - \frac{2\alpha}{\log M}\right).$$

Set the parameter of interest $\theta(P_j) = j$ to be the model index, and distance between parameters to be $\mathbf{dist}(\cdot, \cdot) = \mathbf{1}\{\cdot \neq \cdot\}$, consider $P_j$ to be a product measure of $n$ i.i.d. samples for any $P_j \in \mathcal{M}$, then Lemma H.1 and H.2 under model selection context can be stated as follows:

**Corollary H.3** (Fano's inequality). *For a model family $\mathcal{M}$ contains $M$ many distributions indexed by $j = 1, 2, \ldots, M$ such that $\alpha = \max_{P_j \neq P_k \in \mathcal{M}} \mathbf{KL}(P_j \| P_k)$. If the sample size is bounded as*

$$n \leq \frac{(1 - 2\delta) \log M}{\alpha}\,,$$

*then for any estimator $\widehat{\theta}$ for the model index:*

$$\inf_{\widehat{\theta}} \sup_{j \in [M]} P_j(\widehat{\theta} \neq j) \geq \delta - \frac{\log 2}{\log M}\,.$$

**Corollary H.4** (Tsybakov's method)**.** *For a model family* $\mathcal{M}$ *contains* $M$ *many distributions indexed by* $j = 1, 2, \ldots, M$ *such that* $\alpha = \max_{j \in [M]} \mathbf{KL}(P_j \| P_0)$. *If the sample size is bounded as*

$$n \leq \frac{\log M}{16\alpha},$$

*then for any estimator* $\widehat{\theta}$ *for the model index:*

$$\inf_{\widehat{\theta}} \sup_{j \in [M]} P_j(\widehat{\theta} \neq j) \geq \frac{1}{16}.$$

The model in the context of structure learning is the underlying graph $G$.

# I. Experiment details

We describe the experiment details of Section 6 and provide additional results in this appendix.

**Graph generation**  For our experiments, we simulate poly-forests by initializing an empty adjacency matrix, then randomly ordering the nodes. For each node along the ordering (except the first), an edge is added from a random preceding node in the ordering with probability 80%, ensuring acyclicity and forming a directed forest.

**Gaussian distribution**  We simulate random Gaussian poly-forests according to the following structural equation model:

$$X_k = \beta_k \times X_{\mathrm{pa}(k)} + \eta_k, \quad \eta_k \sim \mathcal{N}(0, \sigma_k^2),$$

where the coefficients are sampled as $\beta_k \sim Unif([-0.5, -0.1] \cup [0.1, 0.5])$, and the noise variances are fixed at $\sigma_k^2 \equiv 1$. Under the Gaussian assumption, we test conditional independence using partial correlations (7). We set the cutoff to be 0.05.

**Bernoulli distribution**  For the synthetic Bernoulli data, root nodes are sampled independently from a $Bern(0.5)$. For each non-root node $X_k$, its conditional distribution given its parents $X_{\mathrm{pa}(k)}$ is given as follows. Let $b_k \sim Unif(l, u)$ and $R_k \sim Unif\{-1, 1\}$, and define the conditional probabilities by:

$$X_k \,|\, X_{\mathrm{pa}(k)} = 1 \sim Bern(0.5 + R_k \times b_k)$$
$$X_k \,|\, X_{\mathrm{pa}(k)} = 0 \sim Bern(0.5 - R_k \times b_k).$$

This construction introduces parent-dependent dependence while ensuring the conditional probabilities remain within a valid range. We set $l = 0.3, u = 0.48$ in our experiments. In this Bernoulli case, we employ the test (3) with the cutoff being 0.05.

**Nonparametric continuous distribution**  To generate synthetic nonparametric continuous data, root nodes are drawn from Uniform distribution $U(0, 1)$. Each subsequent node $X_k$ is generated as a weighted sum of nonlinear transformation of the parent and an individual uniform noise:

$$X_k = f_k(X_{\mathrm{pa}(k)}) \times \frac{3}{10} + U_k(0, 1) \times \frac{7}{10}$$

where $U_k(0, 1) \sim Unif(0, 1)$. The transformation functions $f_k(z)$ are chosen uniformly at random for each parent-child link from a predefined set below, including functions with both range and domain being $[0, 1]$. This process introduces nonparametric dependencies.

$$f_k(z) \sim Unif\left\{0.5 \times \left[\sin(2\pi z) + 1\right], z^2, \frac{\log(1+z)}{\log 2}, 0.5 \times \left[\cos(2\pi z) + 1\right]\right\}$$

For continuous data, we first apply a discretization procedure to convert real-valued observations into categorical representations suitable for contingency-table-based analysis. Following the suggestion of Theorem 5.6 in Neykov et al. (2021), we partition the range into a number of bins determined by the smoothness parameter and the sample size. For variables $(X, Y)$,

we use $n^{2/(5s+2)}$ number of bins. For variable $Z$, we use $n^{2s/(5s+2)}$ number of bins. We set $s = 1$ in this experiment. We assign each observation to a discrete bin in a two- or three-way contingency table, respectively.

Once discretized, we apply the U-statistic Conditional Independence (UCI) test (Kim et al., 2024) to assess whether $X \perp\!\!\!\perp Y \mid Z$. The test computes a U-statistic within each stratum defined by a unique bin of the conditioning variable $Z$, and aggregates the results across strata. We apply weighted U-statistic in our empirical experiments. Since UCI is a permutation test, we set the number of permutation to be 199, following the code provided in `https://github.com/ilmunk/UCI`. We again set the cutoff to be 0.05.

**Evaluation**   For each experiment setup, we report the average (over 50 random replications) Structural Hamming Distance (SHD) between the ground truth and our estimated graph skeleton in Figure 1, and Precise Recovery Rate (PRR) in Figure 2. PRR measures the relative percentage of exact recovery of the true graph structure. Our experimental results demonstrate the robust performance of the `PC-tree` algorithm across various data distributions. As illustrated in the provided subplots for Gaussian, Bernoulli, and Nonparametric synthetic data, the SHD/PRR consistently converges towards 0/100% with increasing sample size. Our empirical results support the theoretical guarantees of the `PC-tree` algorithm.

**Experiment setting**   We consider number of nodes $d = [20, 40, 60, 80, 100]$. To demonstrate the convergence of SHD toward zero, we vary the sample size $n$ from 300 to 3000 across all data types (Bernoulli, Gaussian, and nonparametric) in Figure 1, enabling a consistent evaluation of structural accuracy as sample size increases. To examine convergence behavior in PRR plots, we vary the sample size according to the data type: for nonparametric data, sample sizes range from 700 to 2000; for Bernoulli data, from 3000 to 7000; and for Gaussian data, from 1000 to 10000. The difference is for better presentation and comes from the signal contained in each distribution setup.

**Additional experiments**   We conduct experiments to empirically certify the derived theoretical optimality. To achieve this, we consider two exercises:

- *Comparison with baselines*: We incorporate two baseline structure learning methods (GES and Chow-Liu algorithm) under the same experimental setup. We consider the nonparametric case with $d = 60$ below, evaluated using SHD (standard error in parenthesis). Overall, we observe that PC-tree indeed achieves competitive or superior performance, consistent with our theoretical result that PC-tree with a optimal CI test is minimax optimal.

  | $n$ (Sample size) | 700 | 1300 | 2000 | 2500 | 3000 |
  |---|---|---|---|---|---|
  | PC-tree | 12.4 (3.82) | 0.95 (0.92) | 0.05 (0.21) | 0.00 (0.00) | 0.00 (0.00) |
  | GES | 20.6 (3.39) | 9.60 (3.33) | 6.00 (2.12) | 5.50 (2.74) | 4.00 (2.23) |
  | Chow-Liu | 11.8 (2.96) | 1.30 (1.31) | 0.10 (0.44) | 0.00 (0.00) | 0.00 (0.00) |

- *Scaling with $d$*: We verify if the sample complexity derived for PC-tree is tight by conducting an empirical evaluation for the nonparametric case. Specifically, we generate synthetic data with sample size $n$ proportional to $\log d$, then evaluate the Precise Recovery Rate (PRR). We present the result of PRR vs. $n/\log d$ in the following table for various $d$, where rows correspond to different choices of $d$ and columns correspond to values of $n/\log d$. The observed curves for different dimensions shows a strong alignment with each other, implies that the empirical behavior of PC-tree well coincides with the theoretical prediction.

  | $d \diagdown \frac{n}{\log d}$ | 300 | 350 | 400 | 450 | 500 | 550 |
  |---|---|---|---|---|---|---|
  | 20 | 21.3% | 43.6% | 74.3% | 88.2% | 86.9% | 95.1% |
  | 40 | 18.0% | 42.8% | 63.3% | 78.0% | 91.3% | 97.4% |
  | 60 | 20.4% | 37.9% | 67.7% | 84.9% | 86.3% | 92.2% |
  | 80 | 16.2% | 36.5% | 63.8% | 84.5% | 90.2% | 97.9% |
  | 100 | 28.6% | 32.9% | 71.0% | 89.3% | 95.0% | 96.4% |

**Compute resources**   All experiments were conduced on an Intel Core i7-12800H 2.40GHz CPU.

