# OpenReview forum: "Optimal structure learning and conditional independence testing"
_ICML.cc/2026/Conference — ICML 2026 spotlight_

### Official Review · Reviewer_RZ3o · 2026-03-04

**Soundness:** 3
**Presentation:** 4
**Significance:** 3
**Originality:** 3
**Overall Recommendation:** 5
**Confidence:** 1

**Summary:**

This paper considers the problems of poly-forest learning and conditional independence testing (CI testing). In CI testing, the goal is to determine for a distribution $p(X, Y, Z)$ whether $X, Y$ is independent conditioned on $Z$. In poly-forest learning the goal is to learn the structure of a forest $G$ of $d$ nodes given samples from a distribution $p(X_1, \dots, X_d)$ under the assumption that $p$ satisfies $c$-strong tree-faithfulness and the Markov property w.r.t. $G$.
The main contribution is a reduction from learning the structure of a poly-forest to performing CI testing. More concretely, they show that given an optimal test $\psi$ for doing CI testing, then (under some assumptions) running the PC-tree algorithm gives the optimal sample complexity of learning the corresponding poly-tree structure.

The paper then provides three examples of how their result can be used to construct optimal sample complexity structure learners by instead making optimal CI tests. Here they cover Bernoulli distributions, Gaussian distributions and nonparametric models.

Finally, the paper provides experiments which empirically demonstrate the applicability of their theory.

**Compliance With Llm Reviewing Policy:**

Affirmed.

**Final Justification:**

As I mention in the original review, I am not familiar with this area, and therefore I didn't put weight on significance and originality, since I wouldn't be able to properly assess these qualities.

My concerns in the review was mainly about including a proof sketch of the main theorem and extending the experiments to certify the optimality of their reduction. Both of these points were addressed in the rebuttal, and as such I have updated my recommendation to be accept rather than weak accept.

**Key Questions For Authors:**

1. Did the experiments reflect that the sample complexity of the reduction is optimal?
2. Is it possible that a reduction in the opposite direction exist? That is, given an algorithm that can learn the structure of poly-forests, can you construct a CI tester?

**Limitations:**

yes

**Strengths And Weaknesses:**

**Strengths**\
The introduction and preliminaries of the paper carefully presents the setup of both CI testing and structure learning in a nice manner. It is clear that the authors don't want to assume too much prior knowledge from the readers.
Furthermore, the examples provided in section 4 and 5 nicely complements the result of the paper. Presentation-wise, I think they do a great job.

The paper also provides proofs of all theoretical claims made in the paper (although I did not verify any of these). Furthermore, they provide experiment as a supplement to the theory.

**Weaknesses**\
While the experiments demonstrate the algorithm/reduction performing better when given more samples, I'm having a hard time figuring out if these also demonstrate the optimality of the reduction. I think the experiments would benefit a lot from being compared to a sub-optimal structure learner (but still close to optimal), to demonstrate the point of the reduction giving optimal sample complexity.

Considering that the proof of the main theorem in the paper only takes up ~1.5 pages in the appendix, it would have been nice if a brief proof overview was provided in the main text of the paper.

Since I am not familiar with this area, I don't feel qualified to assess the significance or originality of the work.

Finally, I have 3 very minor things I stumbled on while reading the paper all of which should be easy to fix:
- 110(left): citation should be outside parenthesis.
- 218(right): Could you mention explicitly that $\widehat{G}$ is the estimator in this context.
- 172(right), 297(left), 318(left): Equation overlaps with margin.

---

> ### Author Rebuttal · Authors · 2026-03-31
>
> **[Proof sketch]** Thanks for the nice suggestion, we provide a proof sketch below and will add it to the manuscript. Given a $\mathcal{C}(\mathcal{P}, m)$ with an optimal test $\psi$. Consider $\mathcal{F}(\mathcal{P}, m, c)$ to derive the optimal sample complexity.
>
> For the upper bound, we first construct an amplified version of the test $\psi$ via median trick (Motwani, 1995), which translates the constant Type I and II error guarantees into an exponentially decaying error probability (Proposition C.2). We then apply this amplified test to PC-tree algorithm. The desired upper bound follows by adapting the analysis of Theorem 4.3 in Wang et al. (2024), noting the error probability of each CI test is controlled to ensure uniform consistency over all edge decisions.
>
> For the lower bound, we apply Tsybakov’s method (Corollary G.4). We construct an ensemble of $\asymp d$ candidate distributions, each corresponding to a distinct poly-forest, by embedding the hard instances $p_0$ and $p_1$ into three-node subgraphs and stacking them together. The lower bound follows by verifying the pairwise KL divergences between these distributions are uniformly bounded.
>
> **[Link between theory and experimental results]** We thank the reviewer for the insightful question. We agree it would be valuable to verify whether the empirical results align with the derived theoretical rates. To this end, we *certify* the optimality in the following two ways:
> - *Comparison with baselines*: We add structure learning baselines into the experiments. See the next item (**[Comparison with baselines]**).
> - *Scaling with $d$*: We verify if the sample complexity derived for PC-tree is tight by conducting an empirical evaluation for the nonparametric case. Specifically, we generate data with sample size $n$ proportional to $\log d$, then evaluate Precise Recovery Rate (PRR). We present the result of PRR vs. $n / \log d$ in the following table for various $d$, where rows correspond to different choices of $d$ and columns correspond to values of $n/\log d$. The observed curves for different dimensions show a strong alignment with each other, implies the empirical behavior of PC-tree well coincides with the theoretical prediction.
> |$d$ &nbsp; $\mathbf{\backslash}$ &nbsp; $\frac{n}{\log d}$|300|350|400|450|500|550|
> |:-:|:-:|:-:|:-:|:-:|:-:|:-:|
> |20|21.3%|43.6%|74.3%| 88.2%|86.9%|95.1%|
> |40|18.0%|42.8%|63.3%| 78.0%|91.3%|97.4%|
> |60|20.4%|37.9%|67.7%| 84.9%|86.3%|92.2%|
> |80|16.2%|36.5%|63.8%| 84.5%|90.2%|97.9%|
> |100|28.6%|32.9%|71.0%| 89.3%|95.0%|96.4%|
>
> **[Comparison with baselines]** Thank you for the great suggestion. We agree that the experiments would be more informative with comparisons to additional structure learning baselines. Accordingly, we incorporate two baseline methods (GES and Chow-Liu algorithm) under the same experimental setup. We present the nonparametric case with $d=60$ below, evaluated using SHD (smaller value indicates better performance, standard error in parenthesis). Overall, we observe that PC-tree indeed achieves competitive or superior performance, consistent with our theoretical optimality result.
> |$n$ (Sample size)|700|1300|2000|2500|3000|
> |:-:|:-:|:-:|:-:|:-:|:-:|
> |PC-tree|12.4 (3.82)|0.95 (0.92)|0.05 (0.21)|0.00 (0.00)|0.00 (0.00)|
> |GES|20.6 (3.39)|9.60 (3.33)|6.00 (2.12)|5.50 (2.74)|4.00 (2.23)|
> |Chow-Liu|11.8 (2.96)|1.30 (1.31)| 0.10 (0.44)|0.00 (0.00)|0.00 (0.00)|
>
> **[Reduction in the other direction]** We thank the reviewer for the insightful question. It is not straightforward whether a reduction in the reverse direction can be established, i.e., whether an optimal poly-forest learner can be used to construct an optimal CI tester. Conceptually, CI testing is a local problem defined on a triplet of variables, whereas structure learning is a global problem that aims to recover the entire graph. In this sense, structure learning requires strictly more information than CI testing, which is precisely why we start this work on showing optimality of the local problem (CI testing) implies optimality of the global problem (structure learning).
>
> In the reverse direction, one can obtain a upper bound easily: if the graph structure can be consistently recovered, then the CI relations for any triplet can be read off from the learned graph, thereby yielding a valid CI tester. However, this upper bound is potentially inefficient, as it may require solving a strictly harder global problem in order to answer a local testing problem. The key question, therefore, is whether such a reroute is necessary or achieves the optimality for CI testing. Establishing a tight reduction in this direction would require showing that no loss is incurred when passing from a global estimation problem to a local testing problem, which is not obvious. We agree that this is an interesting theoretical question to investigate in future work.
>
> **[Minor issue]** Thanks for the careful reading. We will fix the suggested issues in the manuscript.

---

> > ### Author Rebuttal · Reviewer_RZ3o · 2026-04-02
> >
> > I thank the authors for addressing all the points in my review. I don't have any further questions, and believe all my points have been addressed satisfactorily. I will update my score to reflect this.

---

> > > ### Author Response · Authors · 2026-04-04
> > >
> > > Thank you very much for your careful reading and for the positive update.
> > > We are glad that our rebuttal addressed your concerns.
> > > We appreciate your time and feedback, which helped improve the clarity of the paper.

---

### Official Review · Reviewer_WmJc · 2026-03-05

**Soundness:** 2
**Presentation:** 2
**Significance:** 2
**Originality:** 2
**Overall Recommendation:** 5
**Confidence:** 3

**Summary:**

This paper studies and establishes a connection between the sample complexity required for structure learning and conditional independence testing in graphical models, and specifically, in poly-forests. The main result (Theorem 3.1) shows that, the minimax sample complexity of structure learning in poly-forests over $d$-dimensional distributions is $\log d/c^\alpha$, when the minimax optimal radius of the associated conditional testing task is $n^{-1/\alpha}$, and $c$ is some "signal strength" parameter that plays a role in both the conditional testing and structure learning tasks. The authors then apply this theorem for deriving optimal structure learning rates for specific distributions, namely Bernoulli, Gaussian and non-parametric continuous distributions. For this, the authors show that the PC algorithm from prior work, instantiated with a suitable conditional independence test that satisfies the conditions of the theorem, achieves this rate. Finally, the authors perform some experiments on synthetic data for these distributions, and show that the error rates for the PC algorithm instantiated with the suggested tests decay as the number of samples increases.

**Compliance With Llm Reviewing Policy:**

Affirmed.

**Final Justification:**

I had some confusions about the theorem statements, and had some concerns about the experiments. Because of this, my initial score was 3 - Weak Reject.

The authors response addresses my concerns. I had maintained that the results in the paper are qualitatively interesting. I have hence increased my score to 5 - Accept

**Key Questions For Authors:**

1) I am quite confused by how the parameter $c$ features in the derived result. After Definition 2.3 for $\mathcal{C}(\mathcal{P},m)$, the authors state, for each sample size $n$, $c=c_n$ is the minimax optimal testing radius, such that if we substitute $c=c_n$ in the definition of $\mathcal{H}_1$ in the hypothesis testing task, then there exists some test whose errors are bounded by 1/10. Now, there is also a $c$ in Definition 2.4 for $\mathcal{F}(\mathcal{P}, m, c)$, by way of measuring $c$-strong-tree-faithfulness, and also for plugging into $\mathcal{H}_1$. Then, Theorem 3.1 states that, given a conditional independence test $\psi$ for $\mathcal{C}(\mathcal{P},m)$ achieving minimax optimal radius $c_n=n^{-1/\alpha}$, the optimal sample complexity for $\mathcal{F}(\mathcal{P}, m, c)$ is $n=\frac{\log(d)}{c^\alpha}$. What is the $c$ in the denominator here? Is it the minimax testing radius of $\psi$, i.e., $c=c_n=n^{-1/\alpha}$? In that case, we get $n=n \log d$ which makes no sense to me.

2)  In Theorem 3.1, what do you mean by "if there exist hard instances $p_0 \in \mathcal{H}_0$ and $p_1 \in \mathcal{H}_1$"? Why the use of the word "hard"? Also, what happens if there do not exist such hard instances? Do you not get any characterization of the optimal sample complexity of the structure learning task in that case? Is it clear that such hard instances always exist/exist often?

3) Is there any relation of the performed experiment to the derived sample complexity bound? If not, what should a reader quantitatively infer from the plots that have been shown? Is there any other experiment which could quantitatively illustrate the derived bounds?

4) Without any other benchmark method to compare to, it is hard to know how well the proposed method fares empirically. Given that the authors claim optimality, is there any other method in the literature for poly-forest learning that the authors could compare to in their experiments?

**Limitations:**

Yes

**Strengths And Weaknesses:**

### Strengths

The tight connection between sample complexities of structure learning and conditional independence testing is qualitatively interesting. It is also nice that the derived result unifies previous optimal structure learning results which were derived for specific classes of distributions under a common framework, and that the algorithm achieving the optimal rates is the simple PC algorithm from prior literature.

### Weaknesses

There are a lot of parameters involved in each of the definitions which are hard to keep track of, and the dependencies have not been clearly stated. The phrasing of Theorem 3.1, which is the main result, is quite unnatural (See questions below). Also, with regards to experiments, beyond qualitatively showing that error rates of the proposed algorithm go down, there isn't any further quantitative insight (in terms of whether the loss plot has any relation to the sample complexity bound derived), nor is there any benchmark method to compare with.

---

> ### Author Rebuttal · Authors · 2026-03-31
>
> **[Parameters in problem definition]** We acknowledge the confusion in how these concepts were introduced. In short, the parameter $c$ is overloaded to denote two different things: 1) The minimax testing radius in CI testing and 2) The faithfulness parameter in structure learning (SL). These are distinct, but linked in our result. To clarify this, we will change the notation $c=c_n$ in Defn 2.3 to $r=r_n$ (for "radius"), and leave $c$ as-is in Defn 2.4.
>
> With this change, our main result connects the two problems: If the minimax testing radius in CI testing is $r_n=(n^{\text{CI}})^{-1/\alpha}$, then the optimal sample complexity for poly-forest learning problem is $n_c^{\text{SL}}=(\log d)/c^\alpha$, where superscripts emphasize the sample sizes are different quantities in the two problems. Specifically,
> - *CI testing problem* The sample size $n^{\text{CI}}$ is fixed, and the goal is to study the minimax testing radius, the smallest dependence under the alternative $r=r_n$ for which the hypotheses can be reliably distinguished. Thus $r_n$ is a function of $n$, in our case $r_n=(n^{\text{CI}})^{-1/\alpha}$. This formulation is standard in minimax testing literature (Canonne at al, 2018; Neykov et al, 2021).
> - *SL problem* The signal strength $c$ is fixed, via the faithfulness assumption (i.e. all non-zero conditional dependence are at least $c>0$). The goal is to study the optimal sample complexity, the smallest sample size $n^{\text{SL}}$ required for reliable graph recovery, which is thus a function of $c$, in our case $n^{\text{SL}}_c=(\log d)/c^{\alpha}$. The notion of optimal sample complexity is common in SL literature (Misra et al., 2020; Wang et al., 2024).
>
> In summary, the distinction lies in which parameter is fixed and which is being studied/optimized. CI testing fixes $n^{\text{CI}}$ and studies the required signal strength $r_n$; while SL fixes $c$ and studies the required sample size $n^{\text{SL}}_c$. These roles can be inverted if desired, however, we chose to present them in the usually way. Afterall, the ultimate goal is the same: To characterize the statistical limit of a problem.
>
> **[Hard instance condition in Thm 3.1]** Thanks for the clarifying question. By "hard instances", we mean a pair of distributions $p_0\in\mathcal{H}_0$ and $p_1\in\mathcal{H}_1$ in CI testing that are close under a suitable divergence (e.g. KL or TV), yet belong to different hypotheses. The term "hard" means such instances are statistically hard to distinguish, i.e. require a large sample size. The precise definition of "hard instance" is given in Thm C.1, which is a formal statement of the main theorem.
>
> This is mostly a matter of terminology: The use of such instances is standard in minimax analysis, and arises in any statistical problem with known optimality, including CI testing (Canonne et al, 2018; Neykov et al, 2021). Their existence is typically implicit, as they capture the fundamental difficulty of the problem. The technique is widely applied in statistics (e.g. Tsybakov, 2008, Thm 2.2; Wainwright, 2019, Eq 15.14). They enable tight lower bound to match the upper bound. For the standard models we study (Bernoulli, Gaussian, nonparametric), such instances are known to exist and can be explicitly constructed (Sec 4–5).
>
> We present the main theorem informally for accessibility to broad audience. While the concepts are standard in minimax community, they may be less familiar in graphical models. To bridge these areas, we chose to begin with an intuitive overview in the main text, with the full statement and conditions on “hard instances” given in Thm C.1 in appendix.
>
> **[Relation b/w theory and experiment]** Thanks for the insightful question. The current experiments illustrate the consistency of PC-tree with optimal CI tests, and the effect of dimension. We agree it would be valuable to empirically verify if the experiment results align with the derived theory. Due to the character limit, we refer to our response to **Reviewer RZ3o [Link between theory and experimental results]**, which shows how we certify the theoretical optimality.
>
> **[Comparison w/ benchmark]** We agree the experiments would benefit from comparison to other structure learning methods. Due to the character limit, we refer to our response to **Reviewer RZ3o [Comparison with baselines]**, which presents new experiments with benchmarks.
> - [Tsybakov, 2008] A. B. Tsybakov. Introduction to Nonparametric Estimation
> - [Canonne at al, 2018] C. L. Canonne, I. D., D. M. K., A. S. Testing conditional independence of discrete distributions
> - [Wainwright, 2019] M. J. Wainwright. High-dimensional statistics: A non-asymptotic viewpoint
> - [Misra et al, 2020] S. Misra, M. V., A. Y. L. Information theoretic optimal learning of gaussian graphical models
> - [Neykov et al, 2021] M. Neykov, S. B., L. W. Minimax optimal conditional independence testing
> - [Wang et al, 2024] Y. Wang, M. G., W. M. T., B. A., A. B. Optimal estimation of Gaussian (poly)trees

---

> > ### Author Rebuttal · Reviewer_WmJc · 2026-04-04
> >
> > Thank you for the response.
> > I would regard the change of notation to be essential in the updated version, since otherwise, a reader is bound to get very confused. In fact, I would encourage the authors to further include the distinction between the two problems above.
> >
> > Thank you as well for the additional experiments. These put the derived bounds into perspective, and I would really encourage the authors to include these as well in the updated version.
> >
> > Given the response and clarifications, I am happy to update my score to 5 - Accept .

---

> > > ### Author Response · Authors · 2026-04-04
> > >
> > > Thank you for the positive update and helpful suggestions. We are glad that our rebuttal addressed your concerns.
> > >
> > > We will incorporate the notation clarification and clearly distinguish the two problems in the revision. We will also include the additional experiments to better illustrate the theoretical results.
> > >
> > > Thank you again for your feedback.

---

### Official Review · Reviewer_GZZc · 2026-03-09

**Soundness:** 4
**Presentation:** 3
**Significance:** 3
**Originality:** 3
**Overall Recommendation:** 5
**Confidence:** 4

**Summary:**

The paper derives a fundamental connection between CI and structure learning in polyforest graphical models. This provides a framework within which different distributions can be studied and their impact on the sample complexity studied. Three different cases are considered and experimental results included that confirm the shape of the theoretical results.

**Compliance With Llm Reviewing Policy:**

Affirmed.

**Key Questions For Authors:**

None

**Limitations:**

As indicated would be great to extend to DAGs

**Strengths And Weaknesses:**

Strengths:
* developing a general framework connecting CI and structure learning for polyforest learning
* providing minimax optimal rates and an algorithm to realise optimality
* experimental results that verify the form of the bounds
Weaknesses:
* not always clear about the assumptions (eg DAGs / polyforest setting)
* precise link between theory and experimental results not fully explored.

---

> ### Author Rebuttal · Authors · 2026-03-31
>
> **[Extension to general DAGs]**
> We sincerely thank the reviewer for recognizing our contribution.
> Our focus on poly-forests serves as a tractable and meaningful starting point for developing a black-box framework that applies across both parametric and nonparametric settings. We view this as a foundational step towards understanding sample efficiency aspect of structure learning.
> Therefore, an intriguing avenue for future research is to generalize our results from poly-forests to general DAGs. In this broader setting, constraint-based approaches like the PC algorithm remain consistent and are candidates to achieve optimality, as they rely on CI testing as workhorse.
> We expect that a similar statistical relationship between CI testing and structure learning could be established for general DAGs. However, additional structural parameters&mdash;such as the maximum in-degree&mdash;will likely be crucial in determining the optimal sample complexity in these more general cases, and in particular, characterizing the dependence on these parameters via sample complexity lower bounds would be a nontrivial and interesting technical question to study.
> We will add this discussion in the manuscript.
>
> **[Link between theory and experimental results]**
> We thank the reviewer for the insightful suggestion. We agree that it would be valuable to further explore the link between our theoretical results and empirical evidence, particularly by assessing whether the observed behavior aligns with the predicted rates.
> To this end, we could *certify* the optimality in the following two ways:
>
> - *Comparison with baselines*: We add structure learning baselines (GES and Chow-Liu algorithm) into the experiments and compare with PC-tree. We present the nonparametric case with $d=60$ below, evaluated using Structural Hamming Distance (smaller value indicates better performance, standard error in parenthesis). Overall, we observe that PC-tree indeed achieves competitive or superior performance, consistent with our theoretical result that PC-tree with a optimal CI test is minimax optimal.
> | $n$ (Sample size) | 700         | 1300        | 2000        | 2500        | 3000        |
> |:-----------------:|:-----------:|:-----------:|:-----------:|:-----------:|:-----------:|
> | PC-tree           | 12.4 (3.82) | 0.95 (0.92) | 0.05 (0.21) | 0.00 (0.00) | 0.00 (0.00) |
> | GES               | 20.6 (3.39) | 9.60 (3.33) | 6.00 (2.12) | 5.50 (2.74) | 4.00 (2.23) |
> | Chow-Liu          | 11.8 (2.96) | 1.30 (1.31) | 0.10 (0.44) | 0.00 (0.00) | 0.00 (0.00) |
> - *Scaling with $d$*: We verify if the sample complexity derived for PC-tree is tight by conducting an empirical evaluation for the nonparametric case.
>     Specifically, we generate synthetic data with sample size $n$ proportional to $\log d$, then evaluate the Precise Recovery Rate (PRR). We present the result of PRR vs. $n / \log d$ in the following table for various $d$, where rows correspond to different choices of $d$ and columns correspond to values of $n/\log d$. The observed curves for different dimensions shows a strong alignment with each other, implies that the empirical behavior of PC-tree well coincides with the theoretical prediction.
> | $d$ &nbsp; $\mathbf{\backslash}$ &nbsp; $\frac{n}{\log d}$ | 300    | 350    | 400    | 450    | 500    | 550    |
> |:-----:|:------:|:------:|:------:|:------:|:------:|:------:|
> | 20    | 21.3%  | 43.6%  | 74.3%  | 88.2%  | 86.9%  | 95.1%  |
> | 40    | 18.0%  | 42.8%  | 63.3%  | 78.0%  | 91.3%  | 97.4%  |
> | 60    | 20.4%  | 37.9%  | 67.7%  | 84.9%  | 86.3%  | 92.2%  |
> | 80    | 16.2%  | 36.5%  | 63.8%  | 84.5%  | 90.2%  | 97.9%  |
> | 100  | 28.6%  | 32.9%  | 71.0%  | 89.3%  | 95.0%  | 96.4%  |
>
> We will add this discussion and the empirical results in the revised manuscript.

---

### Official Review · Reviewer_Wznr · 2026-03-12

**Soundness:** 3
**Presentation:** 3
**Significance:** 3
**Originality:** 3
**Overall Recommendation:** 5
**Confidence:** 2

**Summary:**

This paper studies a connection between structure learning and Conditional Independence (CI) testing. In particular, they show that the minimax sample complexity of structure learning coincides with minimax rate (testing radius) of CI testing for the class of polyforests which are a rich an expressive class of distributions. Moreover, the optimal sample complexity is achieved by the classic PC algorithm. The authors apply their result to specific distributions and recover as special cases some recent results and also obtain new and sharp bounds. Moreover experimental results support their theoretical ones.

**Compliance With Llm Reviewing Policy:**

Affirmed.

**Key Questions For Authors:**

It would be nice to see more discussion regarding the difficulties getting matching upper and lower bounds for general DAGs.

**Limitations:**

Yes

**Strengths And Weaknesses:**

Strengths:

-The paper tackles an important and basic theoretical question in theoretical machine learning/statistics.

-They discover new results for structure learning by  making some new connection between two fields through lower bounds. Prior connections between these subfields was has been used on the algorithmic side, via upper bounds.

-Experimental validation of theoretical results.

Weaknesses:

-The paper does not tackle the general DAG case.

---

> ### Author Rebuttal · Authors · 2026-03-31
>
> **[Tackle general DAGs]**
> We sincerely thank the reviewer for recognizing our contribution.
> Our focus on poly-forests serves as a tractable and meaningful starting point for developing a black-box framework that applies across both parametric and nonparametric settings. We view this as a foundational step towards understanding sample efficiency aspect of structure learning.
> Therefore, an intriguing avenue for future research is to generalize our results from poly-forests to general DAGs. In this broader setting, constraint-based approaches like the PC algorithm remain consistent and are candidates to achieve optimality, as they rely on CI testing as workhorse.
> We expect that a similar statistical relationship between CI testing and structure learning could be established for general DAGs.
> However, additional structural parameters&mdash;such as the maximum in-degree&mdash;will likely be crucial in determining the optimal sample complexity in these more general cases:
>
> - *Upper bound*: In poly-forests, the relationship between the presence/absence of edges and CI relations is more straightforward compared to that in general DAGs&mdash;each edge corresponds to CI statements involving only triplets of random variables $(X,Y,Z)$.
>     This simplicity allows us to establish our reduction between these two problems in a transparent and accessible way.
>     In contrast, additional structural parameters (e.g. maximum degree) should play a role in general DAGs, which has been observed in literature of upper bound analysis.
> - *Lower bound*: For general DAGs, sharp minimax lower bounds remain an open problem due to more complicated dependencies and the combinatorial nature of the graph structure. In particular, characterizing the dependence on the structural parameters via sample complexity lower bounds would be a nontrivial and interesting technical question to study.
>
> We will add this discussion in the manuscript.

---

> > ### Author Rebuttal · Reviewer_Wznr · 2026-04-02
> >
> > I thank the authors for addressing all the points in my review. I have no further questions.

---

> > > ### Author Response · Authors · 2026-04-04
> > >
> > > Thank you very much for your careful reading. We appreciate your time and feedback.

---

### Decision · Program_Chairs · 2026-04-30

**Decision:**

Accept (spotlight)

**Comment:**

The authors have fully resolved the reviewers' concerns regarding notation, experimental benchmarks, and proof sketches. While the focus is currently limited to poly-forests, the depth of the theoretical connection established warrants acceptance.

Please ensure that the clarified notation, the proof sketch for the main theorem, and the additional baseline experiments provided in the rebuttal are integrated into the final camera-ready version as promised.